# A robust and efficient method for Mendelian randomization with hundreds of genetic variants

Stephen Burgess [1,2]*, Christopher N Foley[1], Elias Allara [2,3], James R Staley[2,4] & Joanna M.M. Howson [2,5,6]

Mendelian randomization (MR) is an epidemiological technique that uses genetic variants to distinguish correlation from causation in observational data. The reliability of a MR investigation depends on the validity of the genetic variants as instrumental variables (IVs). We develop the contamination mixture method, a method for MR with two modalities. First, it identifies groups of genetic variants with similar causal estimates, which may represent distinct mechanisms by which the risk factor influences the outcome. Second, it performs MR robustly and efficiently in the presence of invalid IVs. Compared to other robust methods, it has the lowest mean squared error across a range of realistic scenarios. The method identifies 11 variants associated with increased high-density lipoprotein-cholesterol, decreased triglyceride levels, and decreased coronary heart disease risk that have the same directions of associations with various blood cell traits, suggesting a shared mechanism linking lipids and coronary heart disease risk mediated via platelet aggregation.

[1] MRC Biostatistics Unit, University of Cambridge, Cambridge, UK. [2] BHF Cardiovascular Epidemiology Unit, Department of Public Health and Primary Care, University of Cambridge, Cambridge, UK. [3] NIHR Blood and Transplant Research Unit in Donor Health and Genomics, Department of Public Health and Primary Care, University of Cambridge, Cambridge, UK. [4] MRC Integrative Epidemiology Unit, University of Bristol, Bristol, UK. [5] National Institute for Health Research Cambridge Biomedical Research Centre, University of Cambridge and Cambridge University Hospitals, Cambridge, UK. [6] Novo Nordisk Research Centre Oxford, Innovation Building - Old Road Campus, Roosevelt Drive, Oxford, UK. *email: sb452@medschl.cam.ac.uk

Distinguishing between correlation and causation is a fundamentally important problem when trying to understand disease mechanisms. Mendelian randomisation is an epidemiological approach to assess whether a risk factor has a causal effect on an outcome based on observational data[1,2]. The approach treats genetic variants as proxy measures for clinical interventions on risk factors, using the variants analogously to random assignment in a randomised controlled trial[3]. As an example, genetic variants in the *HMGCR* gene region predispose individuals to lower lifelong levels of low-density lipoprotein (LDL) cholesterol, and are also associated with lower risk of coronary heart disease (CHD)[4]. HMGCR is the target of statins, a class of LDL-cholesterol lowering drugs that are an effective treatment for reducing CHD risk.

Mendelian randomisation relies on genetic variants satisfying the assumptions of an instrumental variable (IV)[5]. A valid IV must be associated with the risk factor of interest in a specific way, such that it is not associated with confounders of the risk factor–outcome association, nor does it affect the outcome directly (but only potentially indirectly via its effect on the risk factor of interest). An IV can be used to estimate the average causal effect of the risk factor on the outcome[6]. If we assume that the effect of the risk factor on the outcome is linear and homogeneous in the population, and similarly for the associations of the IVs with the risk factor and outcome, the estimates from different valid IVs should be similar to each other[7]. If estimates differ substantially, then it is likely that not all the IVs are valid.

Genetic variants represent a fertile source of candidate IVs, particularly for genes that have well-understood functions and specific effects on risk factors[8]. Genetic variants are fixed at conception, providing some immunity to reverse causation (as the genetic variant must temporally precede the outcome) and confounding (as the genetic code cannot be influenced by confounding factors that act after conception). However, there are also several reasons why genetic variants may not be valid IVs: such as pleiotropy (that is, a variant affects risk factors on different causal pathways), linkage disequilibrium with a variant that influences another causal pathway, and population stratification[9–11].

For many complex risk factors, Mendelian randomisation analyses may require multiple genetic variants to have enough power to detect a causal effect. Several approaches for making causal inferences with some invalid instruments have previously been proposed. These include methods that assume that a majority of the candidate instruments are valid IVs[12–14], and those that assume a plurality of the candidate instruments are valid IVs[15,16]. The plurality assumption means that out of all groups of candidate instruments having the same asymptotic causal estimate, the largest group is the group of valid IVs. A similar assumption is made in outlier-removal methods, such as MR-PRESSO, which sequentially removes candidate instruments from the analysis based on a heterogeneity measure until all the remaining variants have similar estimates[17]. Other assumptions have also been made: for example, the MR-Egger method assumes that the distribution of direct effects of candidate instruments on the outcome is independent from the distribution of associations with the risk factor (referred to as the Instrument Strength Independent of Direct Effect – InSIDE – assumption)[18].

Mendelian randomisation can exploit summarised data on genetic associations obtained from genome-wide association studies to link modifiable risk factors to disease outcomes[19,20]. These summarised data, comprising beta-coefficients and standard errors from regression of the trait of interest (either risk factor or outcome) on each genetic variant in turn, have been made publicly available by several large consortia[21]. Several of the above methods use summarised data as inputs, and do not require access to individual-level data.

We here introduce the contamination mixture method as a method for obtaining valid causal inferences with some invalid IVs. Compared to other approaches for robust instrumental variable analysis, we believe our proposal has a number of advantages in giving asymptotically consistent estimates under the 'plurality of valid instruments' assumption, being fully likelihood-based, being computationally scalable to large numbers of candidate instruments, and being implemented using summarised genetic data.

In this paper, we examine the performance of the proposed contamination mixture method in an extensive simulation study with realistic parameters. We show that our method performs well compared to previously proposed methods in terms of bias, Type 1 error rate, and efficiency. We then illustrate the use of the method in an example considering the causal effect of high-density lipoprotein (HDL) cholesterol on CHD risk, demonstrating a bimodal distribution of the variant-specific estimates, as well as to consider the effect of LDL-cholesterol on CHD risk (unimodal distribution), and of body mass index (BMI) on risk of Type 2 diabetes (T2D). We investigate factors that identify a group of genetic variants associated with HDL-cholesterol and triglyceride levels having a strong protective effect on CHD risk, showing that several of these variants have the same directions of associations with various blood cell traits. We then discuss the implications for identifying causal risk factors and mechanisms.

## Results

**Overview of the proposed contamination mixture method.** There are two broad contexts in which the contamination mixture method can be used. First, under the assumption that there is a single causal effect of the risk factor on the outcome, the method can estimate this effect robustly and efficiently even when some genetic variants are not valid IVs. Secondly, the method can identify distinct subgroups of genetic variants having mutually similar causal estimates. If multiple such groups are identified, this suggests that there may be several causal mechanisms associated with the same risk factor that affect the outcome to different degrees.

The contamination mixture method is implemented by constructing a likelihood function based on the variant-specific causal estimates. For each genetic variant, an estimate of the causal effect can be obtained by dividing the genetic association with the outcome by the genetic association with the risk factor; thus the only inputs to the method are the genetic association estimates (beta-coefficients and standard errors). If a genetic variant is strongly associated with the risk factor, then its causal estimate will be approximately normally distributed. If a genetic variant is a valid instrument, then its causal estimate will be normally distributed about the true value of the causal effect. If a genetic variant is not a valid instrument, then its causal estimate will be normally distributed about some other value. We assume that the values estimated by invalid instruments are normally distributed about zero with a large standard deviation. This enables a likelihood function to be specified that is a product of two-component mixture distributions, with one mixture distribution for each variant. The computational time for maximising this likelihood directly is exponential in the number of genetic variants. We use a profile likelihood approach to reduce the computational complexity to be linear in the number of variants (Methods).

Briefly, we consider different values of the causal effect in turn. For each value, we calculate the contribution to the likelihood for each genetic variant as a valid instrument and as an invalid instrument. If the contribution to the likelihood as a valid instrument is greater, then we take the variant's contribution as a

valid instrument; if less, then its contribution is taken as an invalid instrument. This gives us the configuration of valid and invalid instruments that maximises the likelihood for the given value of the causal effect. This is a profile likelihood, a one-dimensional function of the causal effect. The point estimate is then taken as the value of the causal effect that maximises the profile likelihood. A 95% confidence interval is constructed by taking the set of values of the causal effect for which twice the difference between the log-likelihood calculated at the value and at the maximum is less than the 95th percentile of a $\chi^2$ distribution with one degree of freedom. We note that the confidence interval from this approach is not constrained to be symmetric or even a single range of values. A confidence interval consisting of multiple disjoint ranges would occur if there were multiple groups of genetic variants having estimates that are mutually consistent within the group, but different between the groups.

**Comparison with previous methods**. To compare the performance of the contamination mixture method against other robust methods for Mendelian randomisation, we performed a simulation study with a broad range of realistic scenarios. We consider the first context, in which there is a single causal effect of the risk factor on the outcome, to enable comparison with other methods. We simulated data in a two-sample setting (that is, the genetic associations with the risk factor and with the outcome are estimated in separate samples of individuals) under 4 scenarios: (1) all genetic variants are valid IVs, (2) invalid IVs have balanced pleiotropic direct effects on the outcome, (3) invalid IVs have directional pleiotropic direct effects on the outcome, and (4) invalid IVs have directional pleiotropic effects on the outcome via a confounder. In the first three scenarios, the InSIDE assumption is satisfied, while in the fourth it is not. We took 100 genetic variants as candidate instruments, with the number of invalid IVs in Scenarios 2–4 being 20, 40, or 60. We performed several methods: the standard inverse-variance weighted (IVW) method that assumes all genetic variants are valid IVs[22], a weighted median method that assumes that a majority of genetic variants are valid IVs[14], the MR-Egger method[18], a weighted mode-based estimation method that assumes a plurality of genetic variants are valid IVs[16], MR-PRESSO[17], and the proposed contamination mixture method. In all, 10,000 simulated datasets were analysed for each scenario (Methods).

Table 1 shows that when all variants are valid IVs, all methods give unbiased estimates. The most efficient robust method, judged by standard deviation of the causal estimates and empirical power to detect a true effect, is the MR-PRESSO method. This method is similar in efficiency to the IVW method, which is optimally efficient with all valid IVs[23], but biased when one or more genetic variants are invalid IVs. The contamination mixture and weighted median methods are slightly less efficient, while the mode-based method and MR-Egger methods are considerably less efficient. Mean estimates are attenuated towards the null in all methods due to weak instrument bias[24]; attenuation was more severe in the MR-Egger and mode-based methods (Table 1). Type 1 error rates for the contamination mixture method were no different to the expected 5% level than expected due to chance in this scenario and in a range of additional scenarios with all variants being valid IVs (Supplementary Table 1). This provides evidence for the validity of the contamination mixture method.

Table 2 shows that no one method outperforms others in every invalid variant scenario. The MR-Egger method performs well in terms of bias under the null and Type 1 error rate in Scenarios 2 and 3, but has the lowest power to detect a positive effect in these scenarios, and is the most biased method in Scenario 4. The IVW

**Table 1 Comparison of methods when all genetic variants are valid instruments.**

| Method | Scenario 1: all instruments valid | | | |
| | Mean | SD | Mean SE | Power |
| --- | --- | --- | --- | --- |
| Null causal effect: $\theta = 0$ | | | | |
| Inverse-variance weighted | 0.000 | 0.022 | 0.023 | 4.2 |
| MR-Egger | 0.001 | 0.061 | 0.062 | 4.4 |
| Weighted median | 0.000 | 0.028 | 0.033 | 2.1 |
| MR-PRESSO | −0.000 | 0.022 | 0.022 | 5.1 |
| Weighted mode-based | 0.001 | 0.061 | 0.199 | 0.3 |
| Contamination mixture | 0.000 | 0.028 | – | 4.9 |
| Positive causal effect: $\theta = +0.1$ | | | | |
| Inverse-variance weighted | 0.094 | 0.023 | 0.024 | 98.1 |
| MR-Egger | 0.063 | 0.064 | 0.066 | 16.1 |
| Weighted median | 0.091 | 0.030 | 0.035 | 78.8 |
| MR-PRESSO | 0.094 | 0.024 | 0.023 | 98.0 |
| Weighted mode-based estimate | 0.082 | 0.051 | 0.143 | 15.5 |
| Contamination mixture | 0.096 | 0.029 | – | 91.0 |

*MR-PRESSO* Mendelian Randomization Pleiotropy RESidual Sum and Outlier method of Verbanck et al.[17]
All methods tested gave unbiased causal estimates in the simulation scenario when there were no invalid instruments. Mean, standard deviation (SD), mean standard error (mean SE) of estimates and empirical power (%) in simulation study for Scenario 1 (all 100 variants valid instruments)

method performs well in Scenario 2, as the random-effect model is able to capture balanced pleiotropic effects with mean zero. However, it is unable to model other types of pleiotropy. The weighted median method has lower bias than the IVW method and reasonable power to detect a causal effect, but has high Type 1 error rate in Scenarios 3 and 4 even with only 20 invalid instruments. The MR-PRESSO method is the most efficient at detecting a causal effect, but also has high Type 1 error rate in Scenarios 3 and 4 even with only 20 invalid instruments. The weighted mode-based estimation method generally has low bias and low Type 1 error rate inflation with up to 40 invalid instruments, but also has low power to detect a causal effect. The contamination mixture method generally has good properties, with low bias and low Type 1 error rate inflation for up to 40 invalid instruments, but much better power than the mode-based estimation method to detect a causal effect. Performance with 60 invalid instruments is generally poor for all methods. While the contamination mixture method has slightly inflated Type 1 error rates in Scenario 2, robust methods are typically only used when the standard method (that is, the IVW method) suggests a causal effect. Hence this is unlikely to lead to additional false positive findings in practice. Coverage with a positive effect is shown in Supplementary Table 2; results followed a very similar pattern to that for Type 1 error rate under the null.

The mean squared error of each method in scenarios with a null causal effect is shown in Fig. 1. The contamination mixture method clearly dominates other methods in terms of overall performance according to this measure, particularly in scenarios with 40 or 60 invalid instruments.

**Unravelling the effect of HDL-cholesterol on CHD risk**. To consider the causal effect of HDL-cholesterol on CHD risk, we took 86 uncorrelated genetic variants previously associated with HDL-cholesterol at a genome-wide level of significance in the Global Lipids Genetic Consortium (GLGC)[25]. Associations with HDL-cholesterol were estimated in the GLGC based on up to 188,577 individuals of European ancestry, and associations with CHD risk from the CARDIoGRAMplusC4D consortium on up to 60,801 CHD cases and 123,504 controls predominantly of

**Table 2 Comparison of methods with some invalid instruments.**

| Method | 20 invalid variants | | | 40 invalid variants | | | 60 invalid variants | | |
|---|---|---|---|---|---|---|---|---|---|
| | Mean | SD | Power | Mean | SD | Power | Mean | SD | Power |
| Null causal effect: $\theta = 0$ | | | | | | | | | |
| Scenario 2: balanced pleiotropy, InSIDE satisfied | | | | | | | | | |
| Inverse-variance weighted | 0.000 | 0.043 | 5.1 | −0.001 | 0.057 | 5.5 | 0.000 | 0.068 | 5.4 |
| MR-Egger | 0.002 | 0.120 | 5.4 | 0.000 | 0.155 | 5.0 | 0.001 | 0.186 | 5.0 |
| Weighted median | 0.000 | 0.034 | 4.3 | −0.000 | 0.043 | 7.5 | 0.001 | 0.057 | 12.4 |
| MR-PRESSO | −0.000 | 0.030 | 7.8 | −0.000 | 0.043 | 12.4 | 0.003 | 0.060 | 9.5 |
| Weighted MBE | −0.004 | 0.118 | 0.3 | 0.002 | 0.066 | 0.9 | −0.001 | 0.161 | 1.8 |
| Contamination mixture | 0.000 | 0.033 | 6.6 | 0.000 | 0.043 | 9.1 | 0.000 | 0.066 | 15.1 |
| Scenario 3: directional pleiotropy, InSIDE satisfied | | | | | | | | | |
| Inverse-variance weighted | 0.133 | 0.031 | 96.6 | 0.266 | 0.038 | 100.0 | 0.400 | 0.044 | 100.0 |
| MR-Egger | 0.005 | 0.111 | 5.3 | 0.010 | 0.135 | 5.4 | 0.015 | 0.148 | 5.9 |
| Weighted median | 0.050 | 0.033 | 26.6 | 0.129 | 0.044 | 88.3 | 0.274 | 0.071 | 99.9 |
| MR-PRESSO | 0.056 | 0.030 | 50.9 | 0.168 | 0.042 | 99.6 | 0.330 | 0.056 | 100.0 |
| Weighted MBE | 0.008 | 0.068 | 0.3 | 0.028 | 0.109 | 2.1 | 0.086 | 0.156 | 22.3 |
| Contamination mixture | 0.016 | 0.034 | 9.7 | 0.042 | 0.044 | 25.4 | 0.144 | 0.173 | 60.9 |
| Scenario 4: pleiotropy via confounder, InSIDE violated | | | | | | | | | |
| Inverse-variance weighted | 0.118 | 0.040 | 85.6 | 0.213 | 0.041 | 99.6 | 0.289 | 0.040 | 100.0 |
| MR-Egger | 0.280 | 0.121 | 81.1 | 0.400 | 0.110 | 96.1 | 0.457 | 0.100 | 99.2 |
| Weighted median | 0.080 | 0.044 | 56.6 | 0.212 | 0.072 | 98.1 | 0.339 | 0.062 | 100.0 |
| MR-PRESSO | 0.051 | 0.038 | 44.8 | 0.155 | 0.051 | 96.2 | 0.260 | 0.051 | 99.9 |
| Weighted MBE | 0.028 | 0.300 | 0.7 | 0.137 | 0.403 | 15.9 | 0.259 | 1.589 | 33.6 |
| Contamination mixture | 0.010 | 0.035 | 9.1 | 0.034 | 0.059 | 20.2 | 0.159 | 0.159 | 52.3 |
| Positive causal effect: $\theta = +0.1$ | | | | | | | | | |
| Scenario 2: balanced pleiotropy, InSIDE satisfied | | | | | | | | | |
| Inverse-variance weighted | 0.094 | 0.044 | 58.2 | 0.095 | 0.058 | 38.4 | 0.095 | 0.069 | 28.2 |
| MR-Egger | 0.063 | 0.120 | 8.7 | 0.060 | 0.155 | 6.3 | 0.060 | 0.188 | 6.3 |
| Weighted median | 0.091 | 0.036 | 69.1 | 0.092 | 0.045 | 59.9 | 0.092 | 0.060 | 51.5 |
| MR-PRESSO | 0.095 | 0.033 | 89.0 | 0.095 | 0.044 | 73.7 | 0.098 | 0.066 | 58.2 |
| Weighted MBE | 0.088 | 0.254 | 15.7 | 0.084 | 0.112 | 16.2 | 0.090 | 0.315 | 17.3 |
| Contamination mixture | 0.097 | 0.036 | 82.7 | 0.098 | 0.047 | 70.8 | 0.101 | 0.074 | 56.9 |
| Scenario 3: directional pleiotropy, InSIDE satisfied | | | | | | | | | |
| Inverse-variance weighted | 0.228 | 0.032 | 100.0 | 0.362 | 0.039 | 100.0 | 0.495 | 0.045 | 100.0 |
| MR-Egger | 0.069 | 0.115 | 9.7 | 0.075 | 0.137 | 9.0 | 0.078 | 0.150 | 9.0 |
| Weighted median | 0.144 | 0.035 | 98.1 | 0.227 | 0.046 | 100.0 | 0.372 | 0.071 | 100.0 |
| MR-PRESSO | 0.156 | 0.032 | 100.0 | 0.278 | 0.046 | 100.0 | 0.440 | 0.058 | 100.0 |
| Weighted MBE | 0.090 | 0.096 | 25.0 | 0.132 | 0.233 | 44.3 | 0.189 | 0.247 | 69.1 |
| Contamination mixture | 0.114 | 0.036 | 92.5 | 0.146 | 0.050 | 94.4 | 0.305 | 0.228 | 98.1 |
| Scenario 4: pleiotropy via confounder, InSIDE violated | | | | | | | | | |
| Inverse-variance weighted | 0.214 | 0.041 | 100.0 | 0.307 | 0.042 | 100.0 | 0.384 | 0.041 | 100.0 |
| MR-Egger | 0.360 | 0.123 | 91.8 | 0.482 | 0.111 | 98.9 | 0.550 | 0.101 | 99.8 |
| Weighted median | 0.176 | 0.047 | 99.1 | 0.309 | 0.073 | 100.0 | 0.434 | 0.063 | 100.0 |
| MR-PRESSO | 0.149 | 0.042 | 99.0 | 0.256 | 0.054 | 100.0 | 0.361 | 0.051 | 100.0 |
| Weighted MBE | 0.102 | 0.487 | 8.4 | 0.225 | 0.401 | 22.5 | 0.453 | 1.416 | 37.8 |
| Contamination mixture | 0.109 | 0.040 | 88.4 | 0.140 | 0.070 | 86.4 | 0.301 | 0.165 | 91.5 |

No one method outperformed all others in every scenario, but the contamination mixture method had good overall performance across scenarios with up to 40 invalid instruments out of 100. Performance with 60 invalid instruments was generally poor for all methods. Mean, standard deviation (SD) of estimates and empirical power (%) in simulation study for Scenarios 2, 3 and 4. *MR-PRESSO* Mendelian Randomization Pleiotropy RESidual Sum and Outlier method of Verbanck et al.[17]; *MBE* mode-based estimate of Hartwig et al.[16]

European ancestry[26] (Supplementary Table 3). Previous analyses for HDL-cholesterol with these variants using the IVW method indicated a protective causal effect, whereas analyses using robust methods (in particular, weighted median and MR-Egger) suggest that the true effect is null[27]. A null effect has been observed in most trials for CETP inhibitors that raise HDL-cholesterol[28,29]. In one trial a modest protective effect was observed[30], although this may be ascribed to the LDL-cholesterol lowering effect of the drug. As the genetic associations with HDL-cholesterol were estimated in the same dataset in which they were discovered, they may be over-estimated due to winner's curse[31]. However, such bias is typically negligible when genetic variants are robustly associated with the exposure, and genetic associations with the outcome as obtained in an independent dataset.

The contamination mixture method gives a likelihood function that is bimodal (Supplementary Fig. 1), with a point estimate (representing the odds ratio for CHD per 1 standard deviation increase in HDL-cholesterol) of 0.67 for the primary maximum

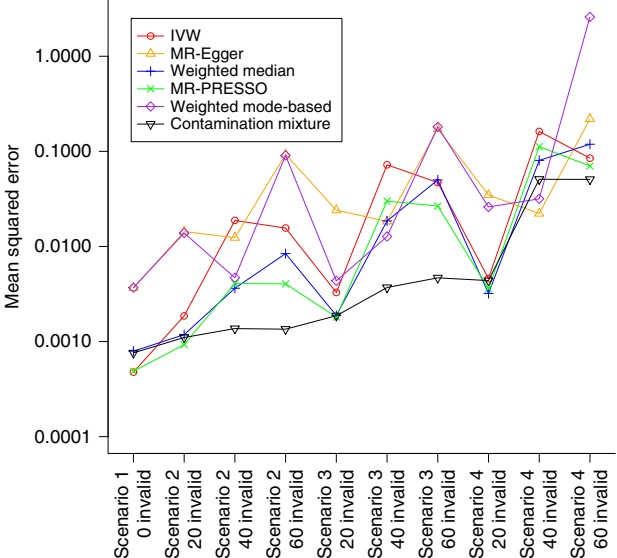

**Fig. 1 Comparison of the methods based on mean squared error criterion.** The mean squared error of the various methods is plotted in each scenario with a null causal effect. The corresponding plot with a positive causal effect was practically identical. The contamination mixture method has the best overall performance according to this measure, particularly in scenarios where 40 or 60 out of the 100 genetic variants were invalid instruments. The vertical axis is plotted on a logarithmic scale.

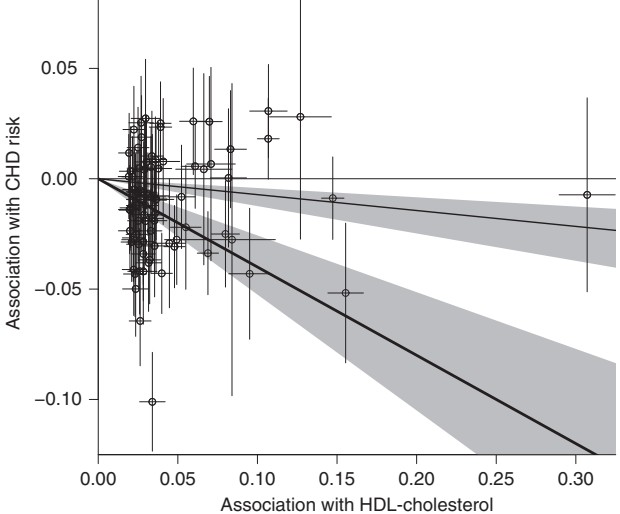

**Fig. 2 Scatter plot of genetic associations.** Genetic associations with HDL-cholesterol (standard deviation units) against genetic associations with CHD risk (log odds ratios). These association estimates are the inputs for the contamination mixture method. Error bars for genetic associations are 95% confidence intervals. Heavy black line is the causal estimate from the contamination mixture method with the strongest signal, lighter black line is the causal estimate from the secondary peak. The grey area is the 95% confidence interval for the causal effect; this comprises two ranges as the likelihood is bimodal.

and 0.93 for the secondary maximum and a 95% confidence interval comprising two disjoint ranges from 0.59 to 0.77, and from 0.88 to 0.96. Estimates are less than one, suggesting a protective effect of HDL-cholesterol on CHD risk. The two regions of the confidence interval suggests the presence of at least two distinct mechanisms by which HDL-cholesterol affects CHD risk, represented by different sets of variants, and the method is uncertain which of the sets is larger. Visual inspection of the scatter graph (Fig. 2) reveals there are several variants suggesting a protective effect of HDL-cholesterol on CHD risk. The bimodal structure of the data would not have been detected by a heterogeneity test. In contrast, a similar analysis for LDL-cholesterol using 75 genome-wide significant variants gives a unimodal likelihood function with a clearly positive causal estimate (Supplementary Fig. 2).

However, 43 of the 86 HDL-cholesterol associated genetic variants are also associated with triglycerides at $p < 10^{-5}$, meaning that the associations with CHD risk may be driven by a harmful effect of triglycerides rather than a protective effect of HDL-cholesterol, as suggested in multivariable Mendelian randomisation analyses[32]. Still, it is worthwhile investigating those variants that evidence the strong protective effect, to see if there is any commonality between them that may suggest a causal mechanism. We proceed to investigate whether there are any traits that preferentially show associations with genetic variants in this cluster as opposed to with variants not in this cluster, as this may help us identify the mechanism driving the genetic associations with lower CHD risk. We note that this investigation is performed without a prior hypothesis, and so should be regarded as an exploratory hypothesis-generating investigation.

We searched for associations of all 86 variants in PhenoScanner[33], a database of summarised data from genome-wide association studies, and found 3209 datasets for which at least one genetic association was available. After restricting to traits for which at least 6 out of the 86 variants were associated at $p < 10^{-5}$, 99 traits remained. For each variant, we calculated the posterior

probability of being a valid genetic variant given a causal effect of 0.67, the estimate from the contamination mixture method. Traits were then ranked according to the mean posterior probability for all variants associated with the trait (Methods, Supplementary Fig. 3). The top ranked trait was platelet distribution width. The next ranked traits were also blood cell traits (Supplementary Table 4): mean corpuscular haemoglobin concentration, and red cell distribution width. This suggests that these variants may be linked to CHD risk relates through blood cell trait-related mechanisms.

We investigated variants that were associated with increased HDL-cholesterol, decreased CHD risk, and at least one of the above blood cell traits (Supplementary Fig. 4). We found 11 genetic variants in 9 distinct gene regions (including those having the largest posterior probability) with a distinct pattern of associations: decreased triglycerides, decreased mean corpuscular haemoglobin concentration, decreased platelet distribution width, and increased red cell distribution width (Fig. 3). This cluster includes a variant in the *LPL* locus. The similarity in the presence and direction of associations with these traits further supports a potential shared causal mechanism. In particular, the finding of platelet distribution width suggests a link with platelet aggregation, a known risk factor for CHD that has previously been linked to HDL-cholesterol[34]. To further investigate evidence of a shared causal mechanism, we performed multi-trait colocalization across these traits for each of the gene regions[35]. For 7 of the regions, there was strong evidence of colocalization for HDL-cholesterol, CHD risk, and at least one of the blood cell traits (Supplementary Table 5). While the claim of a novel causal pathway based on genetic epidemiology alone is premature, this analysis suggests the presence of a mechanism relating lipids to CHD risk that involves platelet aggregation. However, our investigation only highlights these traits as associated with variants in the cluster having a negative association with CHD risk. It does not give any further indication of how the mechanism operates.

| rsid | Nearest gene | HDL-c | TG | LDL-c | CHD | MCHC | PDW | RCDW |
|---|---|---|---|---|---|---|---|---|
| rs4650994 | C1orf220 | 0.021 (0.003) | −0.002 (0.003) | −0.003 (0.004) | −0.029 (0.009) | −0.011 (0.003) | −0.009 (0.004) | 0.017 (0.004) |
| rs4846914 | GALNT2 | 0.048 (0.003) | −0.040 (0.003) | −0.004 (0.004) | −0.031 (0.010) | −0.011 (0.004) | −0.018 (0.004) | 0.030 (0.004) |
| rs7607980 | COBLL1 | 0.045 (0.005) | −0.036 (0.005) | −0.006 (0.006) | −0.029 (0.015) | −0.032 (0.005) | −0.011 (0.006) | 0.019 (0.005) |
| rs9686661 | C5orf67 | 0.028 (0.004) | −0.038 (0.004) | −0.018 (0.005) | −0.042 (0.012) | −0.021 (0.004) | −0.006 (0.004) | 0.006 (0.004) |
| rs2980885 | TRIB1[a] | 0.035 (0.004) | −0.058 (0.004) | −0.031 (0.004) | −0.030 (0.012) | −0.024 (0.004) | −0.010 (0.004) | 0.033 (0.004) |
| rs2954022 | TRIB1[a] | 0.040 (0.003) | −0.078 (0.003) | −0.055 (0.004) | −0.043 (0.009) | −0.031 (0.003) | −0.019 (0.004) | 0.045 (0.004) |
| rs12678919 | LPL | 0.155 (0.006) | −0.170 (0.006) | −0.008 (0.006) | −0.052 (0.016) | −0.030 (0.006) | −0.041 (0.006) | 0.051 (0.006) |
| rs894210 | LPL | 0.069 (0.003) | −0.067 (0.003) | −0.007 (0.004) | −0.034 (0.010) | −0.012 (0.003) | −0.021 (0.004) | 0.021 (0.004) |
| rs10790162 | BUD13[b] | 0.095 (0.007) | −0.230 (0.006) | −0.076 (0.007) | −0.043 (0.015) | −0.038 (0.007) | −0.076 (0.007) | 0.033 (0.007) |
| rs653178 | ATXN2 | 0.026 (0.004) | −0.010 (0.003) | 0.023 (0.004) | −0.064 (0.010) | −0.009 (0.003) | −0.020 (0.004) | 0.009 (0.004) |
| rs2925979 | CMIP | 0.035 (0.004) | −0.020 (0.004) | 0.003 (0.004) | −0.016 (0.010) | −0.012 (0.004) | −0.007 (0.004) | 0.010 (0.004) |

[a]Nearest gene is *RP11-136O12.2*, but signal maps to the *TRIB1* gene.
[b]This gene is adjacent to the *APOA5-APOA4-APOC3-APOA1* cluster on chromosome11.

**Fig. 3 Variants having same directions of associations with blood cell traits.** Details of genetic variants, nearest gene, beta-coefficients (standard errors, SE) for associations with HDL-cholesterol (HDL-c), triglycerides (TG), LDL-cholesterol (LDL-c), coronary heart disease (CHD) risk, mean corpuscular haemoglobin concentration (MCHC), platelet distribution width (PDW), and red cell distribution width (RCDW) for 11 genetic variants in 9 distinct gene regions having a distinct pattern of associations. All associations are orientated to the HDL-cholesterol increasing allele. Red indicates that the association is positive; blue for negative. The brightness of the colouring corresponds to the *p*-value for the strength of association from a *Z*-test; brighter colours correspond to lower *p*-values.

There are several potential reasons why multiple magnitudes of causal effect may be evidenced in the data (Supplementary Fig. 5). HDL-cholesterol is not a single entity. It could be that different size categories of HDL particles influence CHD risk to varying extents[36]. Alternatively, it may be that there are multiple mechanisms by which HDL-cholesterol influences CHD risk, and different genetic variants act as proxies for distinct mechanisms. The identified blood cell traits may act as mediators on one or other of these pathways. Or it could be that the traits are precursors of the risk factor, and some variants influence the traits rather than HDL-cholesterol directly. To assess this, we performed a mediation analysis in which we adjusted for genetic associations with each of the blood cell traits in turn using multivariable Mendelian randomisation[37]. The coefficient for the causal effect of HDL-cholesterol attenuated substantially on adjustment for each of the traits (Supplementary Table 6), and particularly on adjustment for mean corpuscular haemoglobin concentration, suggesting that at least part of the causal effect may be mediated via these blood cell traits. In contrast, associations did not attenuate substantially on adjustment for alternative cardiovascular risk factors (Supplementary Table 6).

**Body mass index and type 2 diabetes risk**. As a further illustration of the ability of this approach to identify biologically relevant pathways, we conducted a similar analysis with BMI as the risk factor and T2D as the outcome. We considered 97 genetic variants previously demonstrated to be associated with body mass index at a genome-wide level of significance[38]. Although the vast majority of genetic variants indicated a harmful effect of BMI on T2D risk, there was a small number of variants suggesting a protective effect (Supplementary Fig. 6). We calculated the posterior probability of having a mild protective effect on T2D risk for all variants, and performed a similar hypothesis-free search of traits in PhenoScanner. The trait that most strongly predicted membership of this subgroup was birth weight. In total, 4 variants were associated with birth weight at $p < 10^{-5}$, including 3 out of the 6 genetic variants suggesting a protective effect of BMI on T2D risk at $p < 0.05$. The hypothesis that high birth weight is protective of T2D (part of the wider Barker hypothesis[39]) has been previously evidenced in Mendelian randomisation analyses[40]. Similarly, evidence from famines such as the Dutch Hongerwinter has suggested that low birthweight and caloric restriction in early childhood is associated with increased risk of

Type 2 diabetes[41]. Our analysis provides further evidence that birth weight is the key factor explaining the opposing effects of BMI on T2D risk: having high BMI from birth appears to be protective of T2D, whereas having high BMI later in life only increases risk of T2D.

## Discussion

In this paper, we have introduced a robust method for Mendelian randomisation analysis referred to as the contamination mixture method. Compared to other robust methods, the contamination mixture method had the best all-round performance in a simulation study – maintaining little bias and close to nominal Type 1 error rates under the null with up to 40% invalid genetic variants, having reasonable power to detect a causal effect in all scenarios, and having the lowest average mean squared error of all methods. In a Mendelian randomisation analysis for HDL-cholesterol on CHD risk, the method detected two separate groups of variants suggesting distinct protective effects on CHD risk. A hypothesis-free search of traits revealed that several variants in the group with the stronger protective effect were associated with platelet distribution width and two other blood cell traits, with consistent directions of association across 11 variants in 9 gene regions. This suggests that the apparent protective effect of these variants may be driven by a shared mechanism, potentially relating to platelet aggregation. This mechanism has some plausibility, as platelets are implicated in atherosclerosis and thrombus formation. The approach also prioritised birth weight as the primary predictor of whether genetic variants that are associated with increased BMI are associated with increased or decreased T2D risk. Overall, the proposed method is able to make reliable causal inferences in Mendelian randomisation investigations with some invalid instruments, as well as highlighting when there are multiple causal effects represented in the data that may be driven by different mechanisms.

Rather than attempting to model the distribution of the estimates from invalid genetic variants, the contamination mixture method proposes that these estimates come from a normal distribution centred at the origin with a wide standard deviation that is pre-specified by the user. We experimented with alternative methods that estimate this distribution. However, as the identity of the invalid instruments is unknown, the performance of these methods was worse than the approach presented here. The runtime of our proposed method is low – on an Intel i7 2.70 GHz processor, the contamination mixture method took 0.08 seconds

to analyse one of the simulated datasets with 100 variants. In comparison, the MR-PRESSO method took 120 seconds, and the mode-based estimation method 81 seconds. The complexity of the contamination mixture model is linear in the number of genetic variants, so the benefit in computational time over other methods would be even greater if more genetic variants were included in the analysis. The previously proposed heterogeneity-penalised method[42], on which the contamination mixture method is based, would be prohibitively slow even with 30 genetic variants. A disadvantage of our proposed method is sensitivity to the standard deviation parameter. In the example of HDL-cholesterol and CHD risk, while the two distinct groups of genetic variants were detected at some values of the standard deviation parameter, at other values distinct groups were not detected.

As genome-wide association studies become larger, the number of genetic variants associated with different traits increases, and it is increasingly unlikely that all these genetic variants are valid instruments for the risk factor of interest. For risk factors with dozens or even hundreds of associated variants, a paradigm shift is required away from approaches that assume the majority of genetic variants are valid instruments, and towards methods that attempt to find genetic variants with similar estimates that might represent a particular causal mechanism. Heterogeneity in variant-specific estimates is inevitable, but heterogeneity should be seen as an opportunity to find causal mechanisms, rather than a barrier to Mendelian randomisation investigations. In our example, we demonstrated how a group of genetic variants having similar causal effects for HDL-cholesterol on CHD risk are linked by their association with platelet distribution width, suggesting a possible mechanism linking lipids to CHD risk related to platelet aggregation. While further research is needed to establish this mechanism, the approach demonstrates the possibility of finding information on causal mechanisms amongst heterogeneous data.

There are many reasons why multiple causal mechanisms may exist between a risk factor and an outcome. It may be that the risk factor is not a single entity, but a compound measurement incorporating multiple risk factors with different causal effects. It may be that the risk factor is a single entity, but there are different ways to intervene on it, leading to different magnitudes of causal effect. Alternatively, it may be that some variants do not affect the risk factor directly, but rather affect a precursor of the risk factor. When multiple causal effects are evidenced in the data, we would not want to label one subset of genetic variants as valid and others as invalid in an absolute sense. Validity of variants is relative to the proposed value of the causal effect – different subsets of variants will be valid for different values of the causal effect. Any cluster of variants may indicate a causal mechanism linking the risk factor to the outcome, even if it is not the largest cluster.

Identifying variables that associate with a cluster of variants having similar causal estimates may help us identify mediators on a particular causal mechanism, or it may help us identify a precursor of the nominal risk factor that is the true causal factor. However, unless we have biological knowledge that a variable identified from such a procedure is on a causal pathway from the risk factor to the outcome, a formal mediation analysis would be speculative. A further possibility is that the link with the variable is coincidental. However, if multiple variants in the cluster are all associated with the same variable with the same direction of association, then a common mechanism is likely.

In conclusion, we have introduced a robust method for Mendelian randomisation that outperforms other methods (including MR-PRESSO) in terms of all-round performance, and can identify groups of variants having similar causal estimates, enabling the identification of causal mechanisms.

## Methods

**Instrumental variable assumptions**. A genetic variant is an instrumental variable if:

1. it is associated with the risk factor of interest,
2. it is not associated with any confounder of the risk factor–outcome association, and
3. it is not associated with the outcome except potentially indirectly via the risk factor[8].

We use the term 'candidate instrumental variable' to indicate a variable that is treated as an instrumental variable, without prejudicing whether it satisfies the instrumental variable assumptions or not.

We assume that the associations between the instrumental variable and risk factor, and the instrumental variable and outcome are linear and homogeneous between individuals with no effect modification, and the causal effect of the risk factor on the outcome is linear[43]. These assumptions are not necessary for the instrumental variable estimate to be a valid test of the causal null hypothesis, but they ensure that all valid instrumental variables estimate the same causal effect.

If summarised data are available representing the association of genetic variant $j$ with the risk factor (beta-coefficient $\hat{\beta}_{Xj}$ and standard error $\text{se}(\hat{\beta}_{Xj})$), and the association with the outcome (beta-coefficient $\hat{\beta}_{Yj}$ and standard error $\text{se}(\hat{\beta}_{Yj})$), then the causal effect of the risk factor on the outcome $\hat{\theta}_j$ can be estimated as:

$$\hat{\theta}_j = \frac{\hat{\beta}_{Yj}}{\hat{\beta}_{Xj}} \tag{1}$$

and its standard error $\text{se}(\hat{\theta}_j)$ as:

$$\text{se}(\hat{\theta}_j) = \frac{\text{se}(\hat{\beta}_{Yj})}{\hat{\beta}_{Xj}}. \tag{2}$$

We refer to $\hat{\theta}_j$ as a variant-specific causal estimate. This formula for the standard error only takes into account the uncertainty in the genetic association with the outcome, but this is typically much greater than the uncertainty in the genetic association with the risk factor (particularly if the recommendation to use only genome-wide significant variants is followed), and does not tend to lead to substantial Type 1 error inflation in practice[22]. In fact, accounting for this uncertainty naively leads to a correlation between the estimated association with the risk factor and the standard error of the variant-specific estimate – which can have more serious consequences in practice than ignoring this source of uncertainty[44,45]. Therefore, while we provide software code so that the user can specify whether to use the simple first-order standard errors or second-order standard errors from the delta method[46], we use the first-order standard errors in our implementation of the method.

**Estimation with multiple instrumental variables**. If there are multiple instrumental variables, then a more precise estimate of the causal effect can be obtained using information on all the instrumental variables. If individual-level data are available, the two-stage least squares method is performed by regressing the risk factor on the instrumental variables, and then regressing the outcome on fitted values of the risk factor from the first stage regression. The same estimate can be obtained by taking a weighted mean of the variant-specific causal estimates using inverse variance weights, as in a meta-analysis[47]. The inverse variance weighted (IVW) estimate can be expressed as:

$$\hat{\theta}_{\text{IVW}} = \frac{\sum_j \hat{\theta}_j \text{se}(\hat{\theta}_j)^{-2}}{\sum_j \text{se}(\hat{\theta}_j)^{-2}}$$
$$= \frac{\sum_j \hat{\beta}_{Yj} \hat{\beta}_{Xj} \text{se}(\hat{\beta}_{Yj})^{-2}}{\sum_j \hat{\beta}_{Xj}^2 \text{se}(\hat{\beta}_{Yj})^{-2}}. \tag{3}$$

The IVW estimate can also be obtained by weighted regression using the following model:

$$\hat{\beta}_{Yj} = \theta\, \hat{\beta}_{Xj} + \epsilon_j, \quad \epsilon_j \sim \mathcal{N}\left(0, \phi^2 \text{se}(\hat{\beta}_{Yj})^2\right) \tag{4}$$

Here, we include an additional term $\phi$, which is the residual standard error in the regression model. In a fixed-effect analysis, this parameter is fixed to be one, but in a random-effects analysis, we allow this term (which represents overdispersion of the variance-specific causal estimates) to be estimated (although we do not allow it to take values less than one, as underdispersion is implausible)[48,49].

The two-stage least squares method (and hence the fixed-effect IVW method) is the most efficient unbiased combination of the variant-specific estimates[23]. However, it is only a consistent estimate of the causal effect if all the candidate instrumental variables are valid. We introduce the contamination mixture method as a robust method for instrumental variable analysis (a robust method is a method that provides asymptotically consistent estimates under weaker assumptions that all genetic variants being valid instruments).

**Contamination mixture method**. Suppose we have $J$ genetic variants that are candidate instrumental variables. We assume that the genetic variant is strongly associated with the risk factor, so that its variant-specific estimate is normally distributed (in practice, we suggest ensuring that each genetic variant is associated with the risk factor at a genome-wide level of significance)[50]. If genetic variant $j$ is a valid instrumental variable, then the variant-specific estimate from that variant $\hat{\theta}_j$ is normally distributed about the true causal parameter $\theta$ with standard deviation equal to the standard error of the ratio estimate $se(\hat{\theta}_j)$:

$$\hat{\theta}_j \sim \mathcal{N}(\theta, se(\hat{\theta}_j)^2) \quad \text{if the } j\text{th genetic variant is a valid instrument.} \quad (5)$$

If the same variant is not a valid instrumental variable, then the ratio estimate from that variant $\hat{\theta}_j$ is normally distributed about some other value $\theta_{F,j}$ with standard deviation equal to the standard error of the ratio estimate $se(\hat{\theta}_j)$. We assume that these $\theta_{F,j}$ are normally distributed about zero with standard deviation $\psi$, such that the distribution of $\hat{\theta}_j$ is:

$$\hat{\theta}_j \sim \mathcal{N}(0, \psi^2 + se(\hat{\theta}_j)^2) \quad \text{if the } j\text{th genetic variant is an invalid instrument.}$$
$$(6)$$

We assume that the standard errors $se(\hat{\theta}_j)$ are fixed and known. Our method makes no attempt to model the distribution of the estimates from invalid instruments. We simply propose a symmetric normal distribution of the estimates about the origin, with the variance of the distribution comprising the proposed variability in the estimands $\theta_{F,j}$ (which is $\psi^2$) plus the uncertainty in the estimate about its asymptotic value (which is the square of its standard error). Although it would be possible to estimate the distribution of the invalid estimands, their distribution is not the focus of the investigation. Our attempts to model their distribution resulted in greater uncertainty in which variants were valid and invalid instruments, and less reliable inferences overall.

If the variant-specific estimate is close to the proposed causal parameter $\theta$, then the likelihood corresponding to the genetic variant being a valid instrument will be larger; if the variant-specific estimate is not close to the proposed causal parameter $\theta$, then the likelihood corresponding to the genetic variant being a invalid instrument will be larger.

We notate the model (that is, the configuration of valid and invalid instruments) as a vector $\zeta$, where $\zeta_j = 1$ when genetic variant $j$ is valid, and $\zeta_j = 0$ otherwise. The likelihood function is then:

$$L(\theta, \zeta) = \prod_j \zeta_j L_{V,j} + (1 - \zeta_j) L_{F,j} \quad (7)$$

$$= \prod_j \zeta_j \times \frac{1}{\sqrt{2\pi se(\hat{\theta}_j)^2}} \exp\left(-\frac{(\theta - \hat{\theta}_j)^2}{2se(\hat{\theta}_j)^2}\right) + \quad (8)$$

$$(1 - \zeta_j) \times \frac{1}{\sqrt{2\pi(\psi^2 + se(\hat{\theta}_j)^2)}} \exp\left(\frac{-\hat{\theta}_j^2}{2(\psi^2 + se(\hat{\theta}_j)^2)}\right),$$

where the likelihood contribution from each genetic variant is $L_{V,j}$ if genetic variant $j$ is valid and $L_{F,j}$ if genetic variant $j$ is invalid. Making inferences using this likelihood is not simple, as the parameter space for $\zeta$ grows exponentially with the number of genetic variants, and there is no guarantee that the likelihood will be unimodal, making it difficult to apply stochastic approaches to explore the model space.

We proceed using a profile likelihood approach. If the causal estimate $\theta$ is fixed, then the optimal model $\zeta$ to maximise the likelihood is clear: we should take $\zeta_j = 1$ if $L_{V,j} > L_{F,j}$ and $\zeta_j = 0$ otherwise. We denote this model choice as $\hat{\zeta}_\theta$. This implies we can easily calculate a profile likelihood $L_p(\theta, \hat{\zeta}_\theta)$ for any value of $\theta$. We perform inferences on $\theta$ by calculating this profile likelihood at a range of values of $\theta$. The causal estimate $\hat{\theta}_p$ is taken as the value of $\theta$ that maximises the profile likelihood, and the confidence interval as the values of $\theta$ such that:

$$2[log(L_p(\hat{\theta}_p, \hat{\zeta}_{\hat{\theta}_p})) - log(L_p(\theta, \hat{\zeta}_\theta))] > \chi^2_{1,0.95} \quad (9)$$

where $\chi^2_{1,0.95}$ is the 95th percentile of a chi-squared distribution with 1 degree of freedom. The confidence interval is based on Wilks' likelihood ratio test, and is not constrained to be symmetric or a single range of values. We note that Wilks' likelihood ratio test is not guaranteed to hold as the number of genetic variants tends towards infinity; as there is one nuisance parameter per genetic variant, and inference for a profile likelihood breaks down as the number of parameters profiled out tends to infinity[51]. However, in practice the number of genetic variants is limited, and no issues with inference were reported in the simulation study. The profile likelihood is a continuous function of $\theta$: it is clearly a continuous function for ranges of $\theta$ when $\hat{\zeta}_\theta$ does not vary, and elements $\zeta_j$ only change their value when the profile likelihood is the same for $\zeta_j = 0$ and $\zeta_j = 1$.

To allow for excess heterogeneity in estimates of the true causal parameter from variants judged to be valid IVs, we estimate an overdispersion parameter as

the residual standard error $\phi$ from weighted regression of the genetic associations with the outcome on the genetic associations with the exposure using the genetic variants judged to be valid at the causal estimate $\hat{\theta}_p$ as in the inverse-variance weighted method. This is analogous to a random-effects model for the IVW method[49]. We then replace $\chi^2_{1,0.95}$ in the above formula with $max(1, \hat{\phi}^2) \times \chi^2_{1,0.95}$. This ensures that variability in the variant-specific causal estimates for the valid IVs beyond what is expected by chance alone results in additional uncertainty in the pooled estimate. However, the point estimate is not changed, nor do we re-evaluate which variants are valid or invalid when overdispersion is present.

We strongly recommend performing a sensitivity analysis for the value of $\psi$. We suggest taking the standard deviation of the ratio estimates based on all the genetic variants multiplied by 1.5 as an initial starting point in considering different values of this parameter. This value was taken in the applied example. This means that the standard deviation of invalid estimands $\theta_{F,j}$ is guided by the variability of the observed ratio estimates, but inflated as the valid instruments will have more similar causal estimates. However, a sensitivity analysis for this parameter is advised. In the applied example, the causal estimate, as well as whether the method detected two separate groups of variants or not, was sensitive to the choice of this parameter (Supplementary Table 7).

As an alternative approach, we considered joint maximisation of the likelihood across both $\psi$ and $\theta$. We re-ran the simulation study for 1000 iterations per invalid instrument scenario with a null causal effect, implementing the original version of the method, and the proposed version jointly maximising the log-likelihood with respect to $\psi$ and $\theta$. Results are shown in Supplementary Table 8. We see that the joint maximisation approach performs less well than the original approach in terms of bias, efficiency, and Type 1 error rate (particularly in Scenario 3). By checking some specific example datasets, we noted that the joint maximisation version often selects a value of $\psi$ that leads to a large group of variants with less similar causal estimates being included in the analysis as valid instruments. This compares with the original version, which excludes more variants from the analysis in these cases, resulting in a lower likelihood, but more robust inferences.

The value of $\psi$ influences which genetic variants are judged to be valid and invalid. Variants are less likely to be judged to be invalid if $\psi$ is too large or too small. If multimodality in the likelihood is detected for any value of $\psi$, then this can be interpreted as evidence for the presence of multiple causal mechanisms. If the causal estimate varies considerably for different values of $\psi$, this suggests that not all genetic variants are valid instruments, and researchers are discouraged from presenting any of these estimates as a single definitive causal estimate. Instead, we encourage researchers to consider whether some variants should be removed from the analysis for being pleiotropic, or to find clusters of variants with similar causal estimates that may represent a coherent causal mechanism.

As stated previously, there are two broad contexts in which the contamination mixture method can be used: to estimate a single causal effect in the presence of invalid IVs, or to identify distinct subgroups of genetic variants having mutually similar causal estimates. The number of subgroups identified by the method is not pre-determined by the analyst, nor is it estimated as a parameter in the model. Rather, for each value of the causal effect, we determine which genetic variants are more likely to be valid instruments, and which invalid. A subgroup is observed when there is a maximum in the likelihood function. This occurs when there are several valid instruments for that value of the causal effect (and hence relatively strong evidence for that value of the causal effect), and relatively less strong evidence for neighbouring values of the causal effect, leading to a distinct peak in the likelihood function. Subgroups could not easily be identified using a heterogeneity measure only, as such an approach would typically downweight or remove from the analysis variants having outlying causal estimates, without assessing whether the estimates from outlying variants were mutually similar.

**Simulation study**. To compare the contamination mixture method with previously developed methods for Mendelian randomisation, we perform a simulation study. We consider four scenarios:

1. no pleiotropy – all genetic variants are valid instruments;
2. balanced pleiotropy – some genetic variants have direct (pleiotropic) effects on the outcome, and these pleiotropic effects are equally likely to be positive as negative;
3. directional pleiotropy – some genetic variants have direct (pleiotropic) effects on the outcome, and these pleiotropic effects are simulated to be positive;
4. pleiotropy via a confounder – some genetic variants have pleiotropic effects on the outcome via a confounder. These pleiotropic effects are correlated with the instrument strength.

In the first three scenarios, the Instrument Strength Independent of Direct Effect (InSIDE) assumption[18] is satisfied; in Scenario 4, it is violated. This is the assumption required for the MR-Egger method to provide consistent estimates.

We simulate data for a risk factor $X$, outcome $Y$, confounder $U$ (assumed unmeasured), and $J$ genetic variants $G_j$, $j = 1, ..., J$. Individuals are indexed by $i$.

The data-generating model for the simulation study is as follows:

$$U_i = \sum_{j=1}^{J} \zeta_j G_{ij} + \epsilon_{Ui}$$

$$X_i = \sum_{j=1}^{J} \gamma_j G_{ij} + U_i + \epsilon_{Xi}$$

$$Y_i = \sum_{j=1}^{J} \alpha_j G_{ij} + \theta X_i + U_i + \epsilon_{Yi} \quad (10)$$

$$G_{ij} \sim \text{Binomial}(2, 0.3) \text{independently for all } j = 1, \dots, J$$

$$\epsilon_{Ui}, \epsilon_{Xi}, \epsilon_{Yi} \sim \mathcal{N}(0,1) \text{independently}$$

$$\gamma_j \sim \text{Uniform}(0.03, 0.1) \text{ independently for all } j = 1, \dots, J$$

The risk factor and outcome are positively correlated due to confounding even when the causal effect $\theta$ is zero through the unmeasured confounder $U$. The genetic variants are modelled as single nucleotide polymorphisms (SNPs) with a minor allele frequency of 30%. A total of $J = 100$ genetic variants are used in each analysis. For each of Scenarios 2–4, we considered cases with 20, 40 and 60 invalid instruments. For valid instruments, the $\alpha_j$ and $\zeta_j$ parameters were set to zero. For invalid instruments, the $\alpha_j$ parameters were either drawn from a uniform distribution on the interval from $-0.1$ to 0.1 (Scenario 2), or from 0 to 0.1 (Scenario 3), or set to zero (Scenario 4). The $\zeta_j$ parameters were either set to zero (Scenarios 2 and 3), or drawn from a uniform distribution on the interval from $-0.1$ to 0.1 (Scenario 4). The causal effect $\theta$ was either set to 0 (no causal effect) or $+0.1$ (positive causal effect). The $\gamma_j$ parameters were drawn from a uniform distribution on 0.03 to 0.1, meaning that the average value of the $R^2$ statistic for the 100 variants across simulated datasets was 9.3% (from 10.5 to 12.7% in Scenario 4) corresponding to an average F statistic of 20.5 (from 23.3 to 28.9 in Scenario 4).

In total, 10,000 datasets were generated in each scenario. We considered a two-sample setting in which genetic associations with the risk factor and outcome were estimated on non-overlapping groups of 20,000 individuals. We compared estimates from the proposed contamination mixture method with those from a variety of methods: the standard IVW method, MR-Egger[18] (both using random-effects), the weighted median method[14], MR-PRESSO[17], and the mode-based estimation (MBE) method of Hartwig et al.[16]. To avoid extreme values in a minority of analyses, we set $\psi = 1$ in the contamination mixture method in all analyses. Each of the methods was implemented using summarised data only. Default values were used by the methods; in particular, the MBE method was implemented using the weighted option with bandwidth $\phi = 1$ under the no measurement error (NOME) assumption (for similarity and thus comparability with other methods), and the MR-PRESSO method was performed trimming variants at a p-value threshold of 0.05 for the heterogeneity test.

For computational reasons (the methods took over 100 times longer to run than the other methods put together), the MR-PRESSO and MBE methods were performed on 1000 datasets per scenario only. Mean squared error was calculated in each scenario by averaging across the 10,000 datasets (1000 datasets for the MR-PRESSO and MBE methods).

**Additional simulation scenarios**. To assess how the contamination mixture method performs with different degrees and directions of confounding, we repeated the simulation study in Scenario 1 with all valid instruments varying the parameters $\delta_X$ and $\delta_Y$ in the equations below:

$$X_i = \sum_{j=1}^{J} \gamma_j G_{ij} + \delta_X U_i + \epsilon_{Xi} \quad (11)$$

$$Y_i = \sum_{j=1}^{J} \alpha_j G_{ij} + \theta X_i + \delta_Y U_i + \epsilon_{Yi}$$

The original simulation study corresponds to $\delta_X = \delta_Y = 1$. We first took $\delta_X = \pm 1$ and $\delta_Y = \pm 1$ and considered a null causal effect $\theta = 0$ and a positive causal effect $\theta = 0.1$. To assess the validity of the method for providing appropriate inferences and appropriately-sized confidence intervals with small samples (for the number of instruments and the number of individuals), we simulated datasets with 10 genetic variants that were valid instruments and a sample size of 5000 for the genetic associations with the risk factor, and 5000 for the genetic associations with the outcome. We also repeated the simulation study in Scenarios 1 to 4 with 40 invalid instruments out of 100. We considered several values for the parameters ($\delta_X, \delta_Y$): (1,1); (1, 0.5); (1, $-0.5$); (0.5, 1); and ($-0.5$, 1). All other parameters were the same as in the original simulation study. We performed 1000 simulations for each set of parameters in each scenario.

Results are shown in Supplementary Tables 1 and 9. With 10 genetic variants that are valid instruments, coverage under the null was no further from the expected 95% level than would be expected by chance alone due to the limited number of simulation replications. With 100 variants, differences between results with different directions and degrees of confounding are somewhat predictable. In Scenarios 1–3, estimates are more precise when there is less variability in the outcome ($\delta_Y$ is smaller in magnitude). In Scenario 3, estimates are also less biased

in this case, as the pleiotropic effects remain constant in magnitude, and so pleiotropic variants can be detected more easily. In Scenario 4, estimates are more biased in this case, as the pleiotropic effects act via the confounder. Additionally in Scenario 4, the direction of bias depends on the direction of confounding. Otherwise, results are no more different between simulations than would be expected by chance alone.

We also repeated the simulation study in Scenarios 2–4 with the genetic effects on the risk factor $\gamma_j$ drawn from a normal distribution with mean 0.065 and standard deviation 0.02. Again, we performed 1000 simulations for each set of parameters in each scenario. Results are shown in Supplementary Table 10. Differences between results when genetic effects were drawn from a uniform and a normal distribution were no more different than would be expected due to chance alone.

**Calculation of the posterior probability**. The posterior probability of a genetic variant being a valid instrumental variable can be calculated as:

$$\pi_{1j} = \frac{\pi_{0j} L(V, j)}{\pi_{0j} L(V, j) + (1 - \pi_{0j}) L(F, j)} \quad (12)$$

where $\pi_{0j}$ is the prior probability of being a valid instrument. We take $\pi_{0j}$ to be the absolute value of the association of variant $j$ with HDL-cholesterol in standard deviation units. This ensures that variants having greater association with the risk factor receive a higher prior weight. If the prior weights for all variants were equal, then a variant having a weak association with the risk factor and an imprecise association with the outcome that is compatible with multiple values of the causal effect could receive the same posterior weight as a variant having a strong association with the risk factor and a precise association with the outcome that is compatible with a far narrow range of values of the causal effect.

Although we could have used the $\zeta_j$ parameters here, these are indicator variables and only take the value zero or one. In contrast, the posterior probabilities account for the value of the variant-specific causal estimate, its precision, and the association of the variant with the risk factor. The combination of these factors is useful in determining which variants have strong evidence for inclusion in a particular subgroup.

**Hypothesis-free search for predictors of subgroup membership**. We consider the subgroup of variants corresponding to the maximum of the likelihood function, and use external data on genetic associations with traits in PhenoScanner to find predictors of subgroup membership. As the external data were not used in determining the subgroups, this can provide validation that the subgroup of variants represents a real biological pathway. For each variant, we calculate the posterior probability of being a valid instrument at the causal estimate. To avoid spurious results, we filter the list of traits to include only traits having at least 6 variants associated at $p < 10^{-5}$. We also exclude traits that are major lipid fractions, and filter out duplicates and highly-related traits. For each trait that remains, we calculate the mean posterior probability for variants associated with the trait at $p < 10^{-5}$.

**Colocalization for HDL-cholesterol associated variants**. Multi-trait colocalization was performed using the Hypothesis Prioritization Colocalization (HyPrColoc) package (https://github.com/jrs95/hyprcoloc). This package performs multi-trait colocalization in a similar way to *moloc*, the multi-trait extension to *coloc*[52], but in a computationally efficient way that allows colocalization of large numbers of traits to be performed. We investigated colocalization between six traits: HDL-cholesterol, triglycerides, CHD risk, mean corpuscular haemoglobin concentration, platelet distribution width, and red cell distribution width. These blood cell traits were selected as variants associated with these traits have the greatest mean posterior probability of belonging to the largest group of variants identified by the contamination mixture method (Supplementary Table 4). Associations with the blood cell traits were estimated in 173,480 unrelated European-descent individuals from the UK Biobank and INTERVAL studies[53]. For each gene region, we took all available variants from the relevant recombination window around the gene[54]. Colocalization was performed using default settings for the priors in the *hyprcoloc* function (prior probability of initial trait association 0.0001, conditional probability of subsequent trait having shared association 0.02), and with the uniform priors setting as the default setting can be overly conservative.

While the exact pattern of colocalization differed between the gene regions, colocalization between HDL-cholesterol, CHD risk, and at least one blood cell trait was observed for 3 of the gene regions using the conservative priors, and for 7 regions using uniform priors (Supplementary Table 5). The posterior probability of colocalization was at least 0.7 in all cases, except when using conservative priors in the *C5orf67* gene region. For this region, there was evidence of colocalization between HDL-cholesterol, triglycerides, CHD risk, and mean corpuscular haemoglobin concentration at posterior probability 0.59, and evidence of colocalization between HDL-cholesterol, triglycerides, and mean corpuscular haemoglobin concentration only (excluding CHD risk) at posterior probability 0.96. For the two gene regions that did not show evidence of colocalization between these traits, one possible explanation is the presence of multiple causal variants in the region; as the *ATXN2* gene region reported colocalization between

HDL-cholesterol and CHD risk, and separately between the blood cell traits. For *COBLL1*, there was colocalization between HDL-cholesterol and the blood cell traits, but not CHD risk.

As this investigation only uses publicly available summarised data on genetic associations with traits and diseases, no specific ethical approval is required.

**Reporting summary**. Further information on research design is available in the Nature Research Reporting Summary linked to this article.

## Data availability
All the data used in this manuscript is publicly available and can be accessed using the PhenoScanner tool at http://www.phenoscanner.medschl.cam.ac.uk/.

## Code availability
Code for the contamination mixture method is presented in the Supplementary Information and is implemented in the MendelianRandomization package (https://cran.r-project.org/web/packages/MendelianRandomization/index.html).

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

## Acknowledgements

This work was funded by the UK Medical Research Council (MR/L003120/1, MC_UU_00002/7), British Heart Foundation (RG/13/13/30194), and the UK National Institute for Health Research Cambridge Biomedical Research Centre. Stephen Burgess is supported by Sir Henry Dale Fellowship jointly funded by the Wellcome Trust and the Royal Society (Grant Number 204623/Z/16/Z). The views expressed are those of the authors and not necessarily those of the National Health Service, the National Institute for Health Research or the Department of Health and Social Care.

## Author contributions

Stephen Burgess developed the initial version of the method, designed the simulation study, analysed the data, and wrote the initial draft of the manuscript. All authors contributed to the development of the method, and to drafting the manuscript.

## Competing interests

While the manuscript was under review, JMMH became a full time employee of Novo Nordisk. The other authors declare no competing interests.
