## [Peer Review File · Nature Communications]

Reviewers' Comments:

Reviewer #1:

Remarks to the Author:

In this report, Burgess et al. propose a new method for Mendelian randomization – an increasingly used technique that seeks causal inference based on genetic variation that is distributed at ~ random in the population.

Although this new technique is likely to be of interest to the general scientific community, I have several suggestions for improvement:

1. This is written more for a genetics audience than a broad Nat Comm audience, efforts should be made to revise text accordingly.
2. It's not clear to me what the input is for the technique – is it just the summary stats for the two traits (HDL and CHD in this example? I think a figure that clearly illustrates what the inputs are would be helpful.
3. For the HDL analysis, were the blood traits selected a priori and used as input? Or did the algorithm select out various clusters of variants and these were then run through PhenoScanner?
4. One additional example could be included to increase generalizability. A straightforward one may be LDL/CHD, where most variants that influence LDL also influence CHD – but there are notable exceptions such as JAK2 p.Val617Phe
5. Can this technique be useful to tease out different pathways relevant for disease biology? For example, look at summary stats from a T2D GWAS and look for clusters of variants that behave similarly and thus might belong to a given pathway?

Reviewer #2:

Remarks to the Author:

This paper proposes a contamination mixture model to capture heterogeneity in Mendelian randomization investigations. The proposed method is used to analyze the potential causal role of HDL cholesterol in the risk of coronary heart disease. The authors claim that they find 11 variants associated with increased HDL, decreased triglyceride, decreased CHD, that had the same directions of associations with some blood cell traits. Overall I find the methodological innovations limited and the biological findings unconvincing, so I cannot recommend publishing this article in a high-impact journal like Nature Communications. Here are some detailed comments:

Modeling assumptions:

1. In the first sentence of the abstract, the authors said "Mendelian randomization (MR) investigations with large number of genetic variants are becoming increasingly common". However, the approximate normal distribution of the variant-specific estimate ($\hat{\theta}_j$) only holds if that genetic instrument is strong enough. Thus this fundamental modeling assumption is questionable when a large number of genetic instruments are used.
2. Relatedly, the "no measurement error" assumption for $\hat{\beta}_{Yj}$ is far from the truth when instruments are weak. This assumption can be avoided by a more careful statistical analysis, Jack Bowden has a recent paper on this. At the very least the authors can use a standard error for $\hat{\theta}_j$ that is computed from the Delta method (which

would have a $se(\hat{\beta}_{X_j})$ term and can be found in any math stats textbook). This standard error is larger than what the authors are using now and will lead to more conservative conclusions.

3. The proposed contamination mixture model is based on the claim that "If there are multiple valid IVs, then the estimates from each IV should be similar to each other" in the second paragraph of the Introduction. However this is not true. Without further assumptions (like effect homogeneity), an instrumental variable only partially identifies the causal effect. So it is entirely possible that valid IVs can give different estimates. See Balke and Pearl (1997) and works by James Robins and Thomas Richardson.

Statistical method:

4. The authors used Wilks' likelihood ratio test to obtain confidence interval of the causal effect. However, here Wilks' test is applied to a profile likelihood here which profiles out a large number of nuisance parameters (γ). Since the number of parameters also grow with the sample size (here the effective sample size is the number of variants), Wilks' theorem does not apply. Further theoretical justification is needed.

5. The authors searched over 3209 traits to find which traits are the HDL variants also associated with. However, no multiple testing procedure is applied to control false positives due to multiple comparisons. This entire applied example looks quite ad hoc. The authors need to give a much more rigorous analysis and show the findings correspond to some real biological mechanism.

6. In the applied example, the authors used the GLGC dataset for instrument selection and statistical inference (this can be immediately spotted from their Figure 2), which leads to selection bias due to winner's curse. In this case, the normality assumption about $\hat{\theta}_j$ is violated.

Reviewer #3:

Remarks to the Author:

Major comments

Burgess et al describe a new method to estimate causal effect when many instruments are invalid. The central idea is smart and the obtained simulation results are very promising. The real data application is interesting and it provides some insight into the mechanism underlying heterogeneous causal effect estimates for the 86 HDL SNPs. Below I list several points that could be considered to improve the paper. In particular, I find the real data application poorly described and difficult to follow.

There are several method simplifications that do not seem to me fully justified. For example, an ad hoc parameter setting in the method "We recommend taking the value of ψ to be the standard deviation of the ratio estimates based on all the genetic variants, multiplied by 1.5." – what is the justification for this? Just empirically tested and it gives the best results for simulations? It seems to be a clear weakness of the method, since it looks like that for the real example, tuning this parameter gives different answers (groups/membership/causal effect estimate?). The choice of ψ is extremely important, since it changes which SNP belongs to the contamination group and that changes the causal effect estimate.

I am puzzled by the choice of the likelihood maximization. Why is it necessary to make a hard cut to set which SNP belongs to which Gaussian component? Why not simply fitting a two component Gaussian mixture, with only the following parameters: proportion of valid instruments, causal effect, ψ (dispersion of valid instruments) and Ψ (dispersion of invalid instruments)? (Membership does not need to be deterministic – only a random effect.) This would be fast, does not require ad hoc estimation of Ψ and no need to have a binary SNP classification. Such method would allow for each SNP to calculate the posterior probability of belonging to the valid or invalid instrument group.

I do not see the reason for allowing dispersion for the valid instruments. The reason for this would exactly be multiple different causal effects, depending on instruments, which would be deemed as invalid instruments. Does it not lead to too many parameters and make model identification difficult?

Is there a second iteration of profile likelihood evaluation once the parameter ψ (dispersion of the valid instruments) is estimated? It may change membership of SNPs and may require re-estimation of the causal effect.

Why is the effect of the confounder U the same in the simulation on both X and Y ? It would be more convincing to try $(U \rightarrow X, U \rightarrow Y)$ effects $(1,1)$, $(1, 0.5)$, $(1, -0.5)$, $(0.5, 1)$, $(-0.5, 1)$ as effect pairs on X and Y respectively.

In the real data application, I'm confused by the description of the results. I could not find how many (and which) of the 86 SNPs were deemed invalid by the method? If I understand well, the authors only talk about the profile likelihood of the causal effect estimate being bimodal. Did the authors seek to find two subgroups among the valid instruments or the bimodal likelihood is caused by the invalid instruments?

I do not see how the new method helps in particular to learn new biology of the HDL->CHD relationship? Any MR method combined with a Cochran's Q test would have identified a non-zero dispersion parameter and one could have followed up what other traits these SNPs are associated to. All the phenoScanner analysis downstream requires little input from the proposed MR method. It seems more like a manual lookup of the effects of the 86 SNPs on other traits.

The link with platelet distribution width is interesting, but the reader is still left without real explanation: is platelet width a confounder of the HDL->CHD correlation/causation? The authors could perform an MR to estimate the causal effect between this variable and HDL/CHD. Is there any mediation happening HDL->trig->CHD or HDL->PDW->CHD?

Minor comments

The SNP effect sizes in the simulations were drawn from uniform, while it is known that it is unrealistic and normal or exponential effect sizes would be far more reasonable.

How was the coverage of the 95% confidence interval under the alternative scenario ($\theta=0.1$)?

If the profile likelihood is bimodal for the real data, shouldn't the authors include 3 component mixture and allowing for direct estimation of two separate causal effects, instead of allowing for heterogeneity (dispersion)?

I don't see the point in calculating the posterior membership for the HDL-associated SNPs. You already estimate γ_j for each SNP j , why not use this classification?

> We would like to express thanks to the reviewers for their time and comments. Replies to points are indicated by angle brackets, and changes to the paper as a result of these comments are clearly indicated. We have numbered the reviewers' points for reference.

> We have made several major changes to the manuscript. In particular, as per the response to A1, we have re-framed the paper towards a more general audience, and as per the response to C6, we have more clearly explained the two contexts for the method: either robust estimation of a single causal effect, or the identification of clusters of variants having similar causal estimates. We have also added two new examples to the manuscript (see A4 and A5): a parallel investigation of LDL-cholesterol and coronary heart disease risk, and an entirely new investigation of genetic variants associated with body mass index and their associations with Type 2 diabetes risk. This shows that the methods proposed in the manuscript are able to provide biological insight in a range of scenarios. <

Reviewers' comments:

Reviewer #1 (Remarks to the Author):

A0. In this report, Burgess et al. propose a new method for Mendelian randomization – an increasingly used technique that seeks causal inference based on genetic variation that is distributed at random in the population.

Although this new technique is likely to be of interest to the general scientific community, I have several suggestions for improvement:

Reply > We thank the reviewer for his/her comments and have taken the suggestions into account in revising the manuscript. <

A1. This is written more for a genetics audience than a broad Nat Comm audience, efforts should be made to revise text accordingly.

Reply > The paper was initially submitted to Nature Genetics and was transferred to Nature Communications without editing. Hence the paper as initially reviewed was written for a genetics audience.

We have edited the framing of the paper for a more general audience. In particular, we introduce the concept of Mendelian randomization in more depth in the introduction, providing a simple example of how the method can be used (page 3). Additionally, we introduce summarized data on genetic associations taken from large GWAS consortia (pages 3-4). While such data are familiar to genetic epidemiologists, they are less familiar to a general audience. We also introduce terminology that may be obscure, such as “two-sample” (page 4). <

A2. It's not clear to me what the input is for the technique – is it just the summary stats for the two traits (HDL and CHD in this example)? I think a figure that clearly illustrates what the inputs are would be helpful.

Reply > We have made it more clear in the text that the summary statistics are the only inputs required for the method: “[methods] do not require access to individual-level data” (page 4) and “thus the only inputs to the method are the genetic association estimates (beta-coefficients and standard errors)” (page 4). The summary statistics (the genetic

associations with HDL-cholesterol and CHD risk) are plotted in Figure 2. We have also clarified this point in the figure legend: “These association estimates are the inputs for the contamination mixture method.” <

A3. For the HDL analysis, were the blood traits selected a priori and used as input? Or did the algorithm select out various clusters of variants and these were then run through PhenoScanner?

Reply > The latter – the Mendelian randomization method provides a log-likelihood function based on the summary associations with the risk factor and outcome only (Supplementary Figure A1). We take causal estimates corresponding to maxima of this function (Figure 2), and identify clusters of variants that are compatible with these values of the causal effect. Then, an algorithm is used to find phenotypic predictors of membership of the dominant cluster.

As the reviewer suggests, an alternative approach would be to use the external data to help form the clusters. This approach could help find tighter clusters. However, it would then be difficult to validate the clusters – as the external data were used to determine the clusters. Our approach has the advantage of validation – if the statistically defined clusters make sense in terms of the external data, this isn’t a self-fulfilling prophecy as the external data weren’t used to determine the clusters.

We have clarified the analysis in response to this comment: “As the external data were not used in determining the clusters, this can provide validation that the subgroup of variants represents a real biological pathway.” (page 24). <

A4. One additional example could be included to increase generalizability. A straightforward one may be LDL/CHD, where most variants that influence LDL also influence CHD – but there are notable exceptions such as JAK2 p.Val617Phe

Reply > We have performed a parallel analysis for LDL cholesterol as we did for HDL cholesterol as suggested. This analysis results in a unimodal log-likelihood function, as is now shown in Supplementary Figure A2.

The variant mentioned by the reviewer has a minor allele frequency in the Exome Aggregation Consortium (ExAC) of 0.00068 (<https://www.ncbi.nlm.nih.gov/clinvar/variation/14662/>), and so does not appear in results from large genome-wide association study consortia. However, while there are isolated variants that associate with increases in LDL-cholesterol and decreases in CHD risk, there is no signal representing a cluster of variants having similar causal estimates, other than the dominant cluster (which suggests a harmful effect of LDL-cholesterol on CHD risk). <

A5. Can this technique be useful to tease out different pathways relevant for disease biology? For example, look at summary stats from a T2D GWAS and look for clusters of variants that behave similarly and thus might belong to a given pathway?

Reply > Yes, one of the novelties of the method is that it can be used to inform disease biology. This has been clarified in the manuscript: “There are two broad settings in which the contamination mixture method can be used.... Secondly, the method can identify distinct subgroups of genetic variants having mutually similar causal estimates. If multiple such groups are identified, this suggests that there may be several causal mechanisms associated

with the same risk factor that affect the outcome to different degrees.” (page 4). Please see also the response to point C6.

> Following the reviewer’s suggestion, we have added the suggested analysis to the paper for body mass index (BMI) and type 2 diabetes (T2D) risk, to see if the approach identifies and validates known pathways relevant for disease risk (page 9). While the majority of variants associated with increased BMI are also associated with increased T2D risk (see scatterplot below), there is a cluster of variants associated with a decrease in T2D risk (Supplementary Figure A5):

We show that the strongest predictor of membership of this cluster according to mean posterior probability is birth weight. This suggests that associations with obesity for variants in this cluster follow a different lifecourse trajectory to other variants. The idea that high birthweight may be a protective factor for Type 2 diabetes is part of the Barker hypothesis (Barker and Osmond, Lancet 1986; doi: 10.1016/s0140-6736(86)91340-1), and has been previously investigated in Mendelian randomization analyses (Warrington et al, Nature Genetics 2019; doi: 10.1038/s41588-019-0403-1), who also demonstrated that genetic predictors of high birthweight are associated with decreased risk of T2D. This finding adds to our understanding of the effect of BMI on T2D risk – interventions that increase an individual’s weight across the life course (such as those on maternal nutrition) are likely to decrease diabetes risk, even if they lead to increases in adult BMI. <

Reviewer #2 (Remarks to the Author):

This paper proposes a contamination mixture model to capture heterogeneity in Mendelian randomization investigations. The proposed method is used to analyze the potential causal role of HDL cholesterol in the risk of coronary heart disease. The authors claim that they find 11 variants associated with increased HDL, decreased triglyceride, decreased CHD, that had the same directions of associations with some blood cell traits. Overall I find the methodological innovations limited and the biological findings unconvincing, so I cannot recommend publishing this article in a high-impact journal like Nature Communications. Here are some detailed comments:

Reply > We are disappointed at the reviewer’s negative view of the paper, which contradicts the view of the other two anonymous reviewers as well as many readers of the paper as a pre-print. We note in passing that this manuscript has been downloaded as a PDF file 517 times from bioRxiv since March 2019, and a twitter thread discussing the paper received positive attention, with 109 ‘likes’ and 56 retweets (<https://twitter.com/stevesphd/status/1102880740323352576>). Hence we believe that there is substantial interest in this method. Nonetheless, we have read through the

reviewer's comments in detail, and edited the paper where we felt the criticism was justified.

With respect to the reviewer's claim of limited methodological innovation, we respectfully disagree. We show through application to HDL and CHD risk, and now also to BMI and T2D risk, that the method can usefully tease apart heterogeneity in genetic associations to find clusters of variants that are connected in a biologically meaningful way. This approach can help to improve understanding of disease aetiology. <

Modeling assumptions:

B1. In the first sentence of the abstract, the authors said "Mendelian randomization (MR) investigations with large number of genetic variants are becoming increasingly common". However, the approximate normal distribution of the variant-specific estimate ($\hat{\theta}_j$) only holds if that genetic instrument is strong enough. Thus this fundamental modeling assumption is questionable when a large number of genetic instruments are used.

Reply > This is an important point, although it relates to the strength of genetic variants, rather than the number of variants in the analysis. We have therefore only used genetic variants that are strongly associated with the risk factor in our applied examples, i.e., variants that are genome-wide significantly associated with the risk factor. For such variants, a normal distribution is a reasonable approximation (please refer to <https://arxiv.org/abs/1512.04486> for a discussion). We do not advocate using instruments that are only weakly associated with the risk factor.

We have clarified in the paper the need to use genetic variants that are strongly associated with the risk factor: "If a genetic variant is strongly associated with the risk factor, then its causal estimate will be approximately normally distributed." (page 4) and "We assume that the genetic variant is strongly associated with the risk factor, so that its variant-specific estimate is normally distributed (in practice, we suggest ensuring that each genetic variant is associated with the risk factor at a genome-wide level of significance)." (page 19). <

B2. Relatedly, the "no measurement error" assumption for $\hat{\beta}_Y$ is far from the truth when instruments are weak. This assumption can be avoided by a more careful statistical analysis, Jack Bowden has a recent paper on this. At the very least the authors can use a standard error for $\hat{\theta}_j$ that is computed from the Delta method (which would have a $se(\hat{\beta}_X)$ term and can be found in any math stats textbook). This standard error is larger than what the authors are using now and will lead to more conservative conclusions.

Reply > We are aware that measurement error is an issue with weak instruments, and so stated our reliance on the uncertainty in the genetic associations with the risk factor being negligible clearly in the Online Methods: "The formula for the standard error only takes into account the uncertainty in the genetic association with the outcome, but this is typically much greater than the uncertainty in the genetic association with the risk factor, and does not tend to lead to Type 1 error inflation [19]." (page 18). We re-iterate (see also point B1) at this point the recommendation to use genome-wide significant variants: "(particularly if the recommendation to use only genome-wide significant variants is followed)" (page 18). This alleviates the problem of measurement error considerably – a genome-wide significant variant ($p < 5 \times 10^{-8}$) corresponds to a beta-coefficient at least 5.5-times its standard error.

Importantly, using the delta method can make inferences less reliable, rather than more reliable. We now state this clearly in the manuscript (page 18): "In fact, accounting for this

uncertainty naively leads to a correlation between the estimated association with the risk factor and the standard error of the variant-specific estimate- which can have more serious consequences in practice than ignoring this source of uncertainty [34, 35].” We therefore have used standard errors calculated from the leading order term of the Delta method only in our analyses. Nevertheless, we have added software code to allow the reader to implement the method using second-order standard errors if desired.

> The use of second order standard errors would not necessarily lead to more conservative conclusions. The research by Jack Bowden shows that the first order standard errors that we use lead to inflation in the Q statistic for assessing the heterogeneity, and hence inflated Type 1 error rates for testing the null hypothesis of homogeneity between the variant-specific (ratio) causal estimates i.e. more likely to falsely reject the null of homogeneity. Importantly, they do not lead to substantial over-precision of the causal estimate, or inflated Type 1 error rates for testing the causal null hypothesis in typical settings, such as those considered in our work. This is clearly supported by our simulations, where we did not observe an increase in type 1 error rate for the causal estimate. In contrast, the second order delta method standard errors lead to under-rejection of the null hypothesis of homogeneity. Paradoxically, (see <https://arxiv.org/abs/1512.04486>), this means that in a random-effects analysis, confidence intervals from the IVW method can be narrower using the second order standard errors – as the estimate of overdispersion is lower. <

B3. The proposed contamination mixture model is based on the claim that "If there are multiple valid IVs, then the estimates from each IV should be similar to each other" in the second paragraph of the Introduction. However this is not true. Without further assumptions (like effect homogeneity), an instrumental variable only partially identifies the causal effect. So it is entirely possible that valid IVs can give different estimates. See Balke and Pearl (1997) and works by James Robins and Thomas Richardson.

Reply > We are aware of this, and stated our reliance on linearity and homogeneity assumptions clearly in the Online Methods: “We assume that the associations between the instrumental variable and risk factor, and the instrumental variable and outcome are linear and homogeneous between individuals with no effect modification, and the causal effect of the risk factor on the outcome is linear [33]. These assumptions are not necessary for the instrumental variable estimate to be a valid test of the causal null hypothesis, but they ensure that all valid instrumental variables estimate the same causal effect.” (page 18).

> We have added reference to these assumptions in the main body of the manuscript: “If we assume that the effect of the risk factor on the outcome is linear and homogeneous in the population, and similarly for the associations of the IVs with the risk factor and outcome, the estimates from different valid IVs should be similar to each other.” (page 4) <

Statistical method:

B4. The authors used Wilks' likelihood ratio test to obtain confidence interval of the causal effect. However, here Wilks' test is applied to a profile likelihood here which profiles out a large number of nuisance parameters (γ). Since the number of parameters also grow with the sample size (here the effective sample size is the number of variants), Wilks' theorem does not apply. Further theoretical justification is needed.

Reply > One would expect that if the method were theoretically flawed, that this would manifest in poor performance, which is not the case for the contamination mixture model. Our method gives reasonable coverage levels in a wide range of simulated examples, and

similar sized confidence intervals to other methods in applied examples. However, we appreciate empirical performance does not prove theoretical correctness.

> The number of profiled out parameters is equal to the number of genetic variants, which remains constant within any given application. While the asymptotic justification of Wilks' theorem requires the sample size (i.e. number of variants) to tend to infinity, we only ever apply the method in the finite sample case, and so the situation that the reviewer describes does not arise.

> We have added explicit caution on this point to the methods (page 20) "We note that Wilks' likelihood ratio test is not guaranteed to hold as the number of genetic variants tends towards infinity; as there is one nuisance parameter per genetic variant, and inference for a profile likelihood breaks down as the number of parameters profiled out tends to infinity [Amemiya 1985]. However, in practice the number of genetic variants is limited, and no issues with inference were reported in the simulation study." and include reference to the relevant chapter on profile likelihood in Advanced Econometrics by Takeshi Amemiya, which discusses Wilks' theorem for profile likelihood. <

B5. The authors searched over 3209 traits to find which traits are the HDL variants also associated with. However, no multiple testing procedure is applied to control false positives due to multiple comparisons. This entire applied example looks quite ad hoc. The authors need to give a much more rigorous analysis and show the findings correspond to some real biological mechanism.

Reply > While we searched through 3209 datasets (we have edited on page 7 to clarify that the search was across 3209 datasets, not 3209 distinct traits), the vast majority of the traits represented were not considered as predictors of cluster membership. To avoid potentially spurious findings (such as traits having a large mean posterior probability due to one or two associated variants), only traits having at least 6 variants associated at $p < 10^{-5}$ were considered. Further, traits were excluded if they represented major lipid classes. Duplicated traits were also filtered to include only the estimates from the largest available dataset. For closely related traits (for example, the list contained dozens of anthropometric measures relating to obesity), a single representative trait was retained. In total, the number of traits that can be considered independent from the perspective of multiple testing is 24.

> To assess the possibility of the finding being a false positive, we applied a bootstrap procedure. For 'Platelet distribution width', 9 genetic variants were associated at $p < 10^{-5}$. The average posterior probability for these variants was 0.187. We drew 9 variants at random from the 86 HDL-associated variants, and calculated the posterior probability for these 9 variants. We repeated this procedure 100,000 times, and recorded the proportion of random draws of 9 variants for which the average posterior probability was greater than 0.187. We did the same for 'Mean corpuscular haemoglobin concentration' (also 9 variants) and 'Red cell distribution width'. The proportions were 0.0014 (PDW), 0.0068 (MCHC), and 0.0058 (RCDW). The Bonferroni corrected p-value is $0.05/24 = 0.002$. This suggests that the finding for platelet distribution width is not simply due to chance. (Additionally, this calculation does not take into account that the variants aren't just associated with the blood cell traits, but that the directions of associations relative to the HDL-cholesterol increasing allele are all consistent.)

> As for the biological mechanism, again, we want to be circumspect in our claims, and avoid being speculative beyond what is reasonable (please see also point C8). Even the strong and consistent genetic associations that we observed here are not sufficient to make a robust

claim of a novel mechanism – this would require experimental work that is well beyond the scope of this paper. However, the finding that blood cell traits relating to blood coagulation is linked with lipids and coronary heart disease is not at all surprising. Coronary artery disease is a result of atherosclerosis – build up of fatty deposits in artery walls. It is entirely plausible that the process of lipid deposition is dependent on the “stickiness” of blood – as suggested by our analysis. We have edited the manuscript to make this more clear (page 8): “While the claim of a novel causal pathway based on genetic epidemiology alone is premature, this analysis suggests the presence of a mechanism relating lipids to CHD risk that involves platelet aggregation.”. <

B6. In the applied example, the authors used the GLGC dataset for instrument selection and statistical inference (this can be immediately spotted from their Figure 2), which leads to selection bias due to winner's curse. In this case, the normality assumption about $\hat{\theta}_j$ is violated.

Reply > We do not believe that this would lead to serious bias in our analysis, as the genetic associations with the outcome are estimated in a separate dataset. As Mendelian randomization principally assesses gene–outcome associations, it is independence of the genetic associations with the outcome that is crucial. Even if there is some overlap between the data sources, provided that the genetic associations with the lipids are estimated in controls only, the two sets of association estimates will be uncorrelated (<https://onlinelibrary.wiley.com/doi/full/10.1002/gepi.21998>). Winner's curse will lead to some of the genetic associations with the risk factor being slightly overestimated, but this shouldn't affect the validity of the analysis.

> To test this, we re-estimated genetic associations with HDL-cholesterol in UK Biobank participants, a completely disjoint sample. The scatterplot of genetic associations is provided below. Results from the contamination mixture method using these association estimates were almost identical to those provided in the paper. <

Reviewer #3 (Remarks to the Author):

C0. Burgess et al describe a new method to estimate causal effect when many instruments are invalid. The central idea is smart and the obtained simulation results are very promising. The real data application is interesting and it provides some insight into the mechanism underlying heterogeneous causal effect estimates for the 86 HDL SNPs. Below I list several points that could be

considered to improve the paper. In particular, I find the real data application poorly described and difficult to follow.

Reply > We thank the reviewer for his/her positive impression of the paper and its contribution to the literature. We note the reviewer's concerns about the presentation of the applied example, and have edited the paper to explain this more clearly as detailed below in response to comments C6, C7, and C8. <

C1. There are several method simplifications that do not seem to me fully justified. For example, an ad hoc parameter setting in the method "We recommend taking the value of ψ to be the standard deviation of the ratio estimates based on all the genetic variants, multiplied by 1.5." – what is the justification for this? Just empirically tested and it gives the best results for simulations? It seems to be a clear weakness of the method, since it looks like that for the real example, tuning this parameter gives different answers (groups/membership/causal effect estimate?). The choice of ψ is extremely important, since it changes which SNP belongs to the contamination group and that changes the causal effect estimate.

Reply > We acknowledge that the dependence of the method on this parameter is not ideal. This is clearly stated in the manuscript as a limitation of the method (page 10). In a simulation study, it is necessary to have a clear rule for determining the value of this parameter, as it is not feasible to consider carefully the value of the parameter for each of thousands of simulated datasets. However, in an applied analysis, researchers can conduct a detailed sensitivity analysis for this parameter. This means that coming up with an algorithmic rule for this parameter is less important in practice.

> We did not intend the "standard deviation multiplied by 1.5" to be anything other than a useful starting point in considering different values of this parameter. We did not intend this to be a prescriptive rule. We appreciate this point was not clear in the original submission.

> We have edited the manuscript to clarify this point: "We strongly recommend performing a sensitivity analysis for the value of ψ . We suggest taking the standard deviation of the ratio estimates based on all the genetic variants multiplied by 1.5 as an initial starting point in considering different values of this parameter." (page 21). We conducted a sensitivity analysis for this parameter in the HDL-CHD analysis (Supplementary Table A5). <

C2. I am puzzled by the choice of the likelihood maximization. Why is it necessary to make a hard cut to set which SNP belongs to which Gaussian component? Why not simply fitting a two component Gaussian mixture, with only the following parameters: proportion of valid instruments, causal effect, ψ (dispersion of valid instruments) and Ψ (dispersion of invalid instruments)? (Membership does not need to be deterministic – only a random effect.) This would be fast, does not require ad hoc estimation of Ψ and no need to have a binary SNP classification. Such method would allow for each SNP to calculate the posterior probability of belonging to the valid or invalid instrument group.

Reply > We agree that there are many ways that this method could have been implemented. There are several reasons why we implemented the method as we did. One is that the number of genetic variants (which is the effective sample size in our analysis) is typically small compared to the sample size in a typical statistical analysis – a sample size in the hundreds is still a "small sample" – and so we are not well-placed to estimate multiple variance parameters, particularly when the cluster membership is uncertain. The reviewer's suggestion sounds similar to the MR-Mix method proposed by Qi and Chatterjee (<https://www.biorxiv.org/content/10.1101/367821v1>), who consider a multi-component

mixture distribution and estimate the parameters suggested by the reviewer in a maximum likelihood framework. This method was demonstrated to have particularly poor performance in a comparison of methods: <https://www.biorxiv.org/content/10.1101/577940v1.abstract>, although its performance did improve slightly when there were 500 genetic variants.

> While we appreciate the reviewer's suggestion, we feel that the method that we have proposed has good performance, outperforming all others in terms of mean squared error in the comparison of methods mentioned above: <https://www.biorxiv.org/content/10.1101/577940v1.abstract>. We are therefore reluctant to make substantial changes to the method. <

C3. I do not see the reason for allowing dispersion for the valid instruments. The reason for this would exactly be multiple different causal effects, depending on instruments, which would be deemed as invalid instruments. Does it not lead to too many parameters and make model identification difficult?

Reply > The reason to allow for dispersion is not to allow for multiple causal effects, but rather is to ensure that the confidence interval is not overly narrow when there is evidence of overdispersion. This is similar to allowing for random-effects when performing a meta-analysis. Overdispersion in this context means that the variability of the variant-specific causal estimates is larger than expected based on the standard errors of the estimates. If this occurs, we allow for the uncertainty in the pooled estimate to be inflated based on the variability in the variant-specific causal estimates. While overdispersion may occur due to invalid instruments, it may also reflect genuine heterogeneity between genetic variants in the way in which they influence the risk factor, and the consequent effect on the outcome. Although we assume that all genetic variants estimate the same causal parameter, we recognize that genetic variants often influence the same risk factor via different mechanisms and to different extents, and so may display heterogeneity in their associations with the outcome. This is reflected in Scenario 2 of the simulation study.

> The difference in interpretation is between excess heterogeneity that is due to the presence of a small number of distinct outliers (which are likely to be invalid instruments and should be removed from the analysis) and excess heterogeneity where the burden of heterogeneity is shared amongst the majority of the variants with no clear outliers (in which case, we estimate a causal effect based on all the variants, but report a wider confidence interval allowing for overdispersion).

> We have clarified this in the manuscript: "This ensures that variability in the variant-specific causal estimates for the valid IVs beyond what is expected by chance alone results in additional uncertainty in the pooled estimate. However, the point estimate is not changed, nor do we re-evaluate which variants are valid or invalid when overdispersion is present." (page 21) <

C4. Is there a second iteration of profile likelihood evaluation once the parameter ψ (dispersion of the valid instruments) is estimated? It may change membership of SNPs and may require re-estimation of the causal effect.

Reply > No, we do not re-evaluate the causal effect once the parameter ψ is estimated. The reason is similar to the preference of Richard Peto for fixed-effects models in meta-analysis and our preference for multiplicative random-effects models in Mendelian randomization:

we do not want detection of heterogeneity to lead to a re-allocation of weights for the variants (typically increasing the weights of outliers), or a re-evaluation of the validity or otherwise of genetic variants (typically including potentially invalid variants in the analysis). We allow for overdispersion to incorporate additional uncertainty in the causal estimate based on valid instrumental variables, not to re-assign variants from invalid to valid instruments.

> We have clarified this in the manuscript (see point C3 above). <

C5. Why is the effect of the confounder U the same in the simulation on both X and Y? It would be more convincing to try (U->X, U->Y) effects (1,1), (1, 0.5), (1, -0.5), (0.5, 1), (-0.5, 1) as effect pairs on X and Y respectively.

Reply > We appreciate the reviewer's point. There is no particular reason why the effect of the confounder should be the same on both X and Y. This model was adopted for simplicity. We chose to prioritize varying other parameters than the effect of the confounder on risk factor and outcome for two main reasons: first, an applied researcher will not typically know U, so these parameters do not have a natural interpretation and the researcher will not know the relevant values of these parameters for his/her application, and second, in our previous investigations, the performance of methods with strong instruments was not particularly sensitive to these parameters.

> We have repeated the simulation study below for a range of values of these parameters as suggested by the reviewer. We considered scenarios with 40 invalid instruments (zero invalid in Scenario 1), and performed the contamination mixture method with a null and a positive causal effect. All other aspects were taken as in the simulation study in the manuscript. We provide these results as Supplementary Table A6.

> While there were differences between results in terms of the degree and direction of bias, these differences are somewhat predictable, and the general pattern of results was unchanged. <

C6. In the real data application, I'm confused by the description of the results. I could not find how many (and which) of the 86 SNPs were deemed invalid by the method? If I understand well, the authors only talk about the profile likelihood of the causal effect estimate being bimodal. Did the authors seek to find two subgroups among the valid instruments or the bimodal likelihood is caused by the invalid instruments?

Reply > We apologize for lack of clarity. Our method has two distinct contexts, which we unfortunately conflated. First, if the analyst believes that there is a single true value of the causal effect, our method allows robust estimation of this quantity. In this context, variants that support the causal estimate with the greatest strength of evidence may be considered "valid" and other variants as "invalid". Second, if the analyst believes that there are potentially multiple causal mechanisms represented in the data that have distinct magnitudes of causal effect, our method can identify these clusters. In this context, we are reluctant to use the words "valid" and "invalid", as the premise of the approach is that there are multiple causal effects represented in the data – it is not that one is "correct" and others are "incorrect". The simulation study mostly takes the first viewpoint and contrasts the proposed method against other methods that estimate a single causal parameter. The applied example of HDL-cholesterol on CHD risk takes the second viewpoint and tries to find clusters of variants with similar causal effect estimates. This point is now clearly delineated

in the manuscript: “There are two broad contexts in which the contamination mixture method can be used....” (page 4) and “[W]e performed a simulation study with a broad range of realistic scenarios. We consider the first context, in which there is a single causal effect of the risk factor on the outcome, to enable comparison with other methods.” (page 4). The additional analysis of LDL-cholesterol on CHD risk is an analysis in the first context, as multiple clusters are not evidenced.

> We only concentrate on the larger subgroup of variants for two reasons: first, the causal effect of this subgroup is larger in magnitude, and so the mechanism is potentially clinically more impactful; and secondly, membership of the larger subgroup is more distinct, with several genetic variants having non-null associations that are clearly compatible with membership of this subgroup. <

C7. I do not see how the new method helps in particular to learn new biology of the HDL->CHD relationship? Any MR method combined with a Cochran’s Q test would have identified a non-zero dispersion parameter and one could have followed up what other traits these SNPs are associated to. All the phenoScanner analysis downstream requires little input from the proposed MR method. It seems more like a manual lookup of the effects of the 86 SNPs on other traits.

Reply > Several aspects of the paper are novel and allow insights to be made. Without the contamination mixture method, it is challenging to decipher that there is a distinct cluster of HDL-cholesterol variants with similar causal estimates – indeed, although the relationship between HDL-cholesterol and CHD risk has been considered many times previously, this cluster of variants has not previously been noticed. Even if an analyst used a heterogeneity test repeatedly, they would not have found this cluster. Rather, each of the members of this cluster would have been removed from the analysis, as they are heterogeneous compared to the majority of variants (which have estimates around zero). Generally speaking, the use of a heterogeneity statistic would not help to discover distinct clusters containing small numbers of variants. The contamination mixture model also provides the posterior weightings that are used in the search through the external data.

> We have clarified in the manuscript the contribution of the method over existing approaches, such as the use of a heterogeneity test as suggested by the reviewer: “The bimodal structure of the data would not have been detected by a heterogeneity test.” (page 7) and “Subgroups could not easily be identified using a heterogeneity measure only, as such an approach would typically downweight or remove from the analysis variants having outlying causal estimates, without assessing whether the estimates from outlying variants were mutually similar.” (page 21). <

C8. The link with platelet distribution width is interesting, but the reader is still left without real explanation: is platelet width a confounder of the HDL->CHD correlation/causation? The authors could perform an MR to estimate the causal effect between this variable and HDL/CHD. Is there any mediation happening HDL->trig->CHD or HDL->PDW->CHD?

Reply > We were deliberately somewhat circumspect as to the interpretation of our finding as we do not believe that strong evidence of a biological mechanism can be made on the strength of observational data alone (please see point B5). However, we were perhaps overly obscure. We have revised our interpretation, and have clarified that we believe the effect on CHD risk arises from some mechanism that influences both lipids and blood cell traits (in particular relating to blood coagulation). We do not believe that there is a

mediation effect, although a formal mediation analysis is not possible as this would require the assumption of a single causal effect of HDL-cholesterol on CHD risk. <

Minor comments

C9. The SNP effect sizes in the simulations were drawn from uniform, while it is known that it is unrealistic and normal or exponential effect sizes would be far more reasonable.

Reply > While we accept the reviewer's criticism, a uniform distribution is convenient for ensuring that genetic effects on the risk factor do not take extreme values, which may lead to anomalous results for individual simulated datasets, and greater variation in the proportion of variance in the risk factor explained by the genetic variants across simulated datasets.

> We repeated with different distributions of the genetic effects, and observed almost identical results – no different than would be expected due to chance alone. We now provide these results in Supplementary Table A7. <

C10. How was the coverage of the 95% confidence interval under the alternative scenario ($\theta=0.1$)?

Reply > We have added this information in a new Supplementary Table A1, which is now referenced in the manuscript. Coverage was very similar with a positive causal effect as with a null causal effect. <

C11. If the profile likelihood is bimodal for the real data, shouldn't the authors include 3 component mixture and allowing for direct estimation of two separate causal effects, instead of allowing for heterogeneity (dispersion)?

Reply > In our algorithm, we vary the causal effect parameter and allow the data to determine the number of maxima. For each value of the causal effect, we determine which genetic variants are more likely to be valid instruments, and which invalid. A maximum is observed when there are several valid instruments for that value of the causal effect (and hence relatively strong evidence for that value of the causal effect), and relatively less strong evidence for neighbouring values of the causal effect, leading to a distinct peak in the log-likelihood function. This approach seems preferable compared with the suggestion of guessing the number of components prior to running the analysis.

> We have clarified in the manuscript that our approach allows the data to determine the number of clusters: "The number of subgroups identified by the method is not pre-determined by the analyst, nor is it estimated as a parameter in the model..." (page 21). <

C12. I don't see the point in calculating the posterior membership for the HDL-associated SNPs. You already estimate γ_j for each SNP j , why not use this classification?

Reply > In revising the manuscript, we realised that we had used the letter γ for the indicator variables denoting the instrument validity and for the genetic associations with the risk factor in the data-generating model. We now use ζ for the indicator variables.

The ζ_j parameters are indicator variables, and as such only take the value zero or one. We have clarified that each posterior weight accounts for the variant-specific causal

estimate (θ_j), its precision, and the genetic association with the risk factor (γ_j). The combination of these factors is useful in determining which variants have strong evidence for inclusion in a particular cluster. This is now clearly stated in the manuscript (page 24). <

Reviewers' Comments:

Reviewer #1:

Remarks to the Author:

I appreciate efforts to respond to my prior comments and note manuscript is substantially improved.

While already strong, any additional efforts to walk the reader through the biologic insights gained by the clustering approach would be appreciated

- e.g. is it really true that if someone is obese from time of birth they are protected against diabetes? that doesn't make intuitive sense to me, nor is it concordant with data based on monogenic obesity mutations -- variants such as those in MC4R increase childhood weight AND type 2 diabetes

-- similarly, is there a specific pathway that is highlighted in the HDL analysis, e.g. LPL pathway that might be worth highlighting?

Reviewer #2:

Remarks to the Author:

I am very disappointed by the authors' response. It seems that the authors basically dismissed my remarks because my "negative view ... contradicts the view of the other two anonymous reviewers as well as many readers of the paper as a pre-print". I read the other reviews, it appears that they are more concerned about the applications while I looked at the methodology more carefully. It is also strange and concerning that the authors use Twitter popularity to suggest the importance of their method and influence the peer review process.

I maintain my previous recommendation to reject this manuscript. The paper sells itself as a novel method that detects interesting and biological sensible heterogeneity. However, I believe the statistical method proposed is still flawed and I am surprised about the authors' reluctance to make any changes that I suggested in the first review, some of them are quite straightforward. For the biological example, I am still unconvinced that multiple comparisons were properly controlled for.

I am frustrated with how the authors repeatedly use inaccurate approximations in their paper (both theoretical derivation and applied example) and suggest that the final method is still fine because each ""approximation" is not too wrong. This includes:

1. Wilks' theorem (which is the main inference method of this paper) simply does not apply because the likelihood in equations (7) and (8) do not contain any element going to infinity (according to authors' reply to my previous comment B4), but Wilks' theorem is asymptotic. It is also entirely unjustified (not even by simulations) to use Wilks' theorem to obtain "confidence intervals" of the "secondary peak" the authors find in Figure 2.

2. The authors' reluctance to move away from the "no measurement error" assumption and simply claim it is not too violated. In particular, I am puzzled by the authors' claim in the reply that "accounting for this uncertainty naively leads to a correlation between the estimated association with the risk factor and the standard error of the variant-specific estimate". In the two-sample MR design, the sampling error in gene-exposure association and the error in gene-outcome association are independent. So where does this spurious correlation come from? There is a comprehensive literature about the distribution of the ratio of two normals, including when it is practically appropriate to use a normal approximation. For example, George Marsaglia (2006) Ratios of Normal Variables. Journal of Statistical Software, 16:4 has some explicit rules for the appropriateness of normal approximation.

3. The authors' is reluctant to acknowledge the potential danger of winner's curse in the paper. In the reply it is indicated that the "slight" overestimation does not change the validity of their method, but this can soon become ugly if the users are unaware of it. Moreover, given that the authors have found another independent dataset for HDL-C (UK Biobank) to show the GLGC estimates are not too biased, I am wondering why the authors don't simply use the UK Biobank in their main analysis so one of their assumptions is exactly satisfied instead of "approximately correct". Will the different dataset still suggest platelet aggregation as a possible explanation for the heterogeneity?

4. The authors' imprecise description of their handling of multiple comparisons. Quoting the rebuttal letter: "To avoid potentially spurious findings, ... only traits having at least 6 variants associated at $p < 10^{-5}$ were considered... In total, the number of traits that can be considered independent from the perspective of multiple testing is 24." Are the "6 variants associated at $p < 10^{-5}$ " among the 86 HDL variants used in this paper, or they could be any variant in the genome? This definitely needs to be clarified. If the authors are referring to the latter, I am surprised that over 99% of the datasets are not powered enough to have at least 6 variants associated at 10^{-5} . If the authors are referring to the former, then I don't understand how 24 is considered as the number of independent traits (because many traits were discarded as they did not have enough significant associations among the 86 HDL variants). This might look sketchy as the "Bootstrap p-value" for platelet distribution computed by the authors is 0.0014, barely passed the Bonferroni threshold $0.05/24=0.002$ after adjusting for 24 "independent" traits.

Additional comments:

5. Figure 2 shows that the confidence interval of the secondary peak does not cover 0. Does that disprove model (6) which says that the invalid instruments are centered around 0?

6. Supplementary Table A1 shows that the coverage of the proposed contamination mixture model is always lower than the nominal 95% level (and may be as low as 30% in unfavorable situations). Why did the authors say "One would expect that if the method were theoretically flawed, that this would manifest in poor performance, which is not the case for the contamination mixture model" in response to my criticism of the theoretical soundness?

7. If like the authors suggested in the abstract, that there is a "shared mechanism linking lipids and CHD risk relating to platelet aggregation", why not simply check this by running another MR study using traits indicating platelet aggregation as the exposure?

8. If I understand it correctly, the authors' "bootstrap p-value" is just a permutation test. Why is it called bootstrap?

Reviewer #3:

Remarks to the Author:

The authors did a good job reassuringly addressing most of my comments. Only three comments require further work in my view.

C1. There are several method simplifications that do not seem to me fully justified. For example, an ad hoc parameter setting in the method "We recommend taking the value of to be the standard deviation of the ratio estimates based on all the genetic variants, multiplied by 1.5." – what is the justification for this? Just empirically tested and it gives the best results for simulations? It seems to be a clear weakness of the method, since it looks like that for the real example, tuning this parameter gives different answers (groups/membership/causal effect estimate?). The choice of Psi is extremely important, since it changes which SNP belongs to the contamination group and that changes the causal effect estimate.

Reply

> We acknowledge that the dependence of the method on this parameter is not ideal. This is clearly stated in the manuscript as a limitation of the method (page 10). In a simulation study, it is necessary to have a clear rule for determining the value of this parameter, as it is not feasible to consider carefully the value of the parameter for each of thousands of simulated datasets. However, in an applied analysis, researchers can conduct a detailed sensitivity analysis for this parameter. This means that coming up with an algorithmic rule for this parameter is less important in practice.

> We did not intend the "standard deviation multiplied by 1.5" to be anything other than a useful starting point in considering different values of this parameter. We did not intend this to be a prescriptive rule. We appreciate this point was not clear in the original submission.

> We have edited the manuscript to clarify this point: "We strongly recommend performing a sensitivity analysis for the value of ψ . We suggest taking the standard deviation of the ratio estimates based on all the genetic variants multiplied by 1.5 as an initial starting point in considering different values of this parameter." (page 21). We conducted a sensitivity

C1R. Thanks for these explanations. Could you give clear guidance what to do if different ψ values lead to different causal effect estimates? Supplementary Table A5 goes into this direction, but the results are rather worrying, different ψ lead to vastly different causal effects – it seems we could get any answer we want by setting ψ accordingly. The major claimed insight in the application was the detection of two distinct slopes for the HDL->CHD relationship, but it is supported only for some limited values of ψ . How can we know which one is correct? This makes me think: why not first generate the profile likelihood over a grid of ψ and θ values? This would mean no tuning is needed for ψ .

C6. In the real data application, I'm confused by the description of the results. I could not find how many (and which) of the 86 SNPs were deemed invalid by the method? If I understand well, the authors only talk about the profile likelihood of the causal effect estimate being bimodal. Did the authors seek to find two subgroups among the valid instruments or the bimodal likelihood is caused by the invalid instruments?

Reply > We apologize for lack of clarity. Our method has two distinct contexts, which we unfortunately conflated

C6R. Thanks for the clarification around the applied example. This allowed me to digest these results. I am still puzzled for several reasons:

a. The authors talk about two local maxima in the profile likelihood 0.67 and 0.93, but the slopes in the Figure are negative and of course the causal effect should be negative. Is there a typo?

b. The definition of valid instrument is quite arbitrary. If for example the true causal effect is zero and there are a set of SNPs showing correlated pleiotropy [<https://www.biorxiv.org/content/10.1101/682237v1>] for another trait, wouldn't the method pick up those pleiotropic SNP as the most likely valid instruments? In such scenario does it make any sense to look for enrichment for trait-associated SNPs among the more valid instruments?

c. I still struggle to understand what the enrichment analysis shows. Despite my concern (see point b) about the meaning of "valid instrument" indicated by your method, I wonder what it means if many of those SNPs that tend to be more valid are enriched to be associated with another trait? Is this other trait (e.g. PDW) a mediator or rather a trait for which these "valid instruments" may exhibit pleiotropy? The latter one would exactly mean that they are invalid, so I

assume you rather mean that these traits are mediators or what else could they be? Why invalid instruments should be less enriched for a potential mediator? Why is the path through "valid instruments" more meaningful than the path through "invalid instruments" (which are mostly invalid because of pleiotropy, which is biologically just as meaningful)?

d. What is described here "We investigated variants that were associated with increased HDL-cholesterol, decreased CHD risk, and at least one of the above blood cell traits" and below has nothing to do with the findings of the proposed method and comes a bit out of the blue to me.

e. It should be noted that only PDW survives Bonferroni correction, the other traits could occur by chance (or by being correlated to PDW).

f. In general, I believe that for the readers it would be extremely useful to show a DAG (including HDL, PDW, CHD and potentially other factors) describing how the presumably valid instruments (higher posterior probability of being valid) are enriched for being associated with platelet distribution width. Testing enrichment without any particular underlying model in mind is less meaningful.

C8. We do not believe that there is a mediation effect, although a formal mediation analysis is not possible as this would require the assumption of a single causal effect of HDL-cholesterol on CHD risk.

C8R. The fact that (under some ψ settings) the data yields evidence for two distinct slopes does not mean that there is no single causal effect, it simply means that there are SNPs that violate an MR assumption (most probably the exchangeability, see "correlated pleiotropy" above). Of course, it could also mean context-dependent causal effects, but the marginal causal effect (integrating out all other factors) is unique. Moreover, stratified mediation analysis could be done by using two groups of instruments, once those supporting the 0.67 causal effect, once the other SNPs leading to 0.93 causal effect.

> We would like to express thanks again to the reviewers for their time and comments. Replies to points are given in brackets, and changes to the paper as a result of these comments are clearly indicated. We have numbered the reviewers' points for reference as A1R (reviewer 1, revision comment 1), A2R, and so on. A1, A2, and so on refer to comments on the original submission.

> We would like to express thanks for the supportive attitude of the editor with respect to reviewer 2. While we understand his/her viewpoint, several of his/her comments prioritize technical correctness over practical utility. Clearly a balance between the two is required, as there are some occasions when simplifying assumptions can be made that improve applicability of the method without impairing performance. Reviewer 3 has done a good job in distinguishing those areas where technical correctness does matter (eg comment B1R) from those areas where standard approximations are reasonable (eg comments B2R and B3R), and in providing constructive suggestions for comments in the first category to assess and improve technical correctness. As per the editor's suggestion, as have focused on the comments from reviewer 2 that reviewer 3 thought were valid (B1R, B4R, B5R and B6R). <

Reviewer #1 (Remarks to the Author):

AOR. I appreciate efforts to respond to my prior comments and note manuscript is substantially improved.

> We appreciate the reviewer's perspective and positive comments on the revision. <

A1Ra. While already strong, any additional efforts to walk the reader through the biologic insights gained by the clustering approach would be appreciated
- e.g. is it really true that if someone is obese from time of birth they are protected against diabetes? that doesn't make intuitive sense to me, nor is it concordant with data based on monogenic obesity mutations -- variants such as those in MC4R increase childhood weight AND type 2 diabetes

> We would be reluctant to use the word "obese" to describe a newborn baby!

> The fetal origins hypothesis (https://en.wikipedia.org/wiki/Fetal_origins_hypothesis) is still a matter of scientific debate, but we are far from the first people to suggest that having increased weight in early childhood may be beneficial for lowering risk of Type 2 diabetes later in life (or conversely, low weight in childhood may be a risk factor for Type 2 diabetes – see also evidence from the Dutch Hungerwinter and other similar famines). One theory is that early exposure to insufficient calories leads to epigenetic changes and the body storing food more readily as fat.

> While the rs489693 variant in the *MC4R* gene region is strongly associated with adult BMI and Type 2 diabetes, and was associated with childhood BMI in the Early Growth Genetics Consortium (<https://www.ncbi.nlm.nih.gov/pubmed/26604143>), it was not associated with birth weight in the UK Biobank (results from Ben Neale, available from <http://www.phenoscanter.medschl.cam.ac.uk/?query=rs489693&catalogue=GWAS&p=1&proxies=None&r2=0.8&build=37>). Hence it does not represent a counter-example to this hypothesis.

> We have added a further reference to the manuscript from the existing literature on the fetal origins hypothesis: "Conversely, evidence from famines such as the Dutch

Hongerwinter has suggested that low birthweight and caloric restriction in early childhood is associated with increased risk of Type 2 diabetes [van Abeelen 2012]." (page 9). <

A1Rb. -- similarly, is there a specific pathway that is highlighted in the HDL analysis, e.g. LPL pathway that might be worth highlighting?

> We have added specific reference in the manuscript that the highlighted cluster of variants includes a variant in the *LPL* locus (page 9). We searched Gene Ontology to see if there was any shared mechanism between the genes that we may have missed, but no pathway was highlighted. <

Reviewer #2 (Remarks to the Author):

B0R. I am very disappointed by the authors' response. It seems that the authors basically dismissed my remarks because my "negative view ... contradicts the view of the other two anonymous reviewers as well as many readers of the paper as a pre-print". I read the other reviews, it appears that they are more concerned about the applications while I looked at the methodology more carefully. It is also strange and concerning that the authors use Twitter popularity to suggest the importance of their method and influence the peer review process.

I maintain my previous recommendation to reject this manuscript. The paper sells itself as a novel method that detects interesting and biological sensible heterogeneity. However, I believe the statistical method proposed is still flawed and I am surprised about the authors' reluctance to make any changes that I suggested in the first review, some of them are quite straightforward. For the biological example, I am still unconvinced that multiple comparisons were properly controlled for.

> Clearly the number of people interested in a paper as a pre-print is no indication of technical correctness. However, this does provide evidence that while the reviewer "find[s] the methodological innovations limited and the biological findings unconvincing", that others disagree. <

I am frustrated with how the authors repeatedly use inaccurate approximations in their paper (both theoretical derivation and applied example) and suggest that the final method is still fine because each ""approximation" is not too wrong. This includes:

B1R. Wilks' theorem (which is the main inference method of this paper) simply does not apply because the likelihood in equations (7) and (8) do not contain any element going to infinity (according to authors' reply to my previous comment B4), but Wilks' theorem is asymptotic. It is also entirely unjustified (not even by simulations) to use Wilks' theorem to obtain "confidence intervals" of the "secondary peak" the authors find in Figure 2.

Reviewer 3's comment on B1R: I agree that there is a conceptual difficulty with the justification of the Wilks' theorem. For an asymptotic property the "n" should go to infinity, but in this application the "n" is the number of instruments, which is not only finite, but also equals to the number of nuisance parameters. I feel that the reviewer and the authors do not seem to talk about the same issue. I am not an expert of this field, but if the number of instruments are kept constant and even if the study sample size increases, it does not ensure convergence in distribution, thus normality is not guaranteed. Still it could be that the convergence is quick and it leads to no issue. This could be tested by bootstrap/jackknife estimates of the estimator variance and could be compared to the approximation provided by the Wilks' theorem.

> Reviewer 2's criticism here is that we use a method that is asymptotically valid for inference when our sample size doesn't really tend to infinity. However, almost all classical methods for statistical inference are only valid asymptotically – even the p-values in logistic regression. And yet logistic regression is regularly carried out when the sample size for the investigation is not infinite.

> However, as clarified by reviewer 3, there is a legitimate criticism here: the sample size in our case is the number of genetic variants, and we make the assumption that inferences from Wilks' theorem are reasonable despite the sample size being small. To assess this, we have conducted a simulation study in which we calculate coverage of the 95% confidence interval. While we appreciate reviewer 3's suggestion to compare the jackknife variance estimate, Wilks' theorem does not provide a variance estimate. Moreover, we believe coverage is the key property of an estimator in practice.

> We consider a simulation set-up with 10 genetic variants and a sample size of 5000 for both the gene—risk factor and gene—outcome associations, and consider scenarios with positive and negative confounding, a null causal effect, and a positive causal effect. The choice of 10 genetic variants was made to demonstrate the performance of the method with a small number of variants. Additionally, a sample size of 5000 is relatively small for a Mendelian randomization investigation. Even so, coverage is 95% in each scenario – no Type 1 inflation is observed.

> This simulation has been added to the Supplementary Material, and is discussed in the main body of the text: "Type 1 error rates for the contamination mixture method were no different to the expected 5% level than expected due to chance in this scenario and in a range of additional scenarios with all variants being valid IVs (Supplementary Table A1). This provides evidence for the validity of the contamination mixture method." (page 5). <

B2R. The authors' reluctance to move away from the "no measurement error" assumption and simply claim it is not too violated. In particular, I am puzzled by the authors' claim in the reply that "accounting for this uncertainty naively leads to a correlation between the estimated association with the risk factor and the standard error of the variant-specific estimate". In the two-sample MR design, the sampling error in gene-exposure association and the error in gene-outcome association are independent. So where does this spurious correlation come from? There is a comprehensive literature about the distribution of the ratio of two normals, including when it is practically appropriate to use a normal approximation. For example, George Marsaglia (2006) Ratios of Normal Variables. Journal of Statistical Software, 16:4 has some explicit rules for the appropriateness of normal approximation.

> As supported by reviewer 3, the no measurement error assumption has "repeatedly shown not to be an issue with well-powered GWAS" (quote from reviewer 3). We therefore maintain that our original response to the reviewer's point was entirely adequate.

B3R. The authors' is reluctant to acknowledge the potential danger of winner's curse in the paper. In the reply it is indicated that the "slight" overestimation does not change the validity of their method, but this can soon become ugly if the users are unaware of it. Moreover, given that the authors have found another independent dataset for HDL-C (UK Biobank) to show the GLGC estimates are not too biased, I am wondering why the authors don't simply use the UK Biobank in their main analysis so one of their assumptions is exactly satisfied instead of "approximately correct". Will the different dataset still suggest platelet aggregation as a possible explanation for the heterogeneity?

> As stated by reviewer 3, winner's curse "has minute impact when the selected instruments are strong". We have added a statement reflecting this to this paper: "As the genetic associations with HDL-cholesterol were estimated in the same dataset in which they were discovered, they may be over-estimated due to winner's curse [Dudbridge and Newcombe 2019]. However, such bias is typically negligible when genetic variants are robustly associated with the exposure and genetic associations with the outcome as obtained in an independent dataset. Identical results were obtained using genetic associations estimated in UK Biobank (results not shown)." (page 6). <

B4R. The authors' imprecise description of their handling of multiple comparisons. Quoting the rebuttal letter: "To avoid potentially spurious findings, ... only traits having at least 6 variants associated at $p < 10^{-5}$ were considered... In total, the number of traits that can be considered independent from the perspective of multiple testing is 24." Are the "6 variants associated at $p < 10^{-5}$ " among the 86 HDL variants used in this paper, or they could be any variant in the genome? This definitely needs to be clarified. If the authors are referring to the latter, I am surprised that over 99% of the datasets are not powered enough to have at least 6 variants associated at 10^{-5} . If the authors are referring to the former, then I don't understand how 24 is considered as the number of independent traits (because many traits were discarded as they did not have enough significant associations among the 86 HDL variants). This might look sketchy as the "Bootstrap p-value" for platelet distribution computed by the authors is 0.0014, barely passed the Bonferroni threshold $0.05/24=0.002$ after adjusting for 24 "independent" traits.

> We have clarified that we consider GWAS datasets where at least 6 out of the 86 variants are associated at $p < 10^{-5}$ (page 8).

> We added these p-values as an attempt to provide a quantitative assessment of whether the genetic associations with the blood cell traits were stronger than expected by chance alone in the revision. While we believe it did provide some supporting evidence of this, the approach was not convincing to either reviewer 2 or reviewer 3 (see point C6R). Additionally, the approach used to calculate bootstrap p-values does not take into account that the genetic associations with blood cell traits were all in the same direction – which is further supportive evidence that the associations represent a biological mechanism.

> As this approach is quite subjective with respect to multiple testing and does not fully represent the strength of evidence, and as the approach was negatively viewed by two reviewers, we have now omitted it from the manuscript. We maintain our stance that the fact that 11 genetic variants in 9 gene regions have concordant directions of association with lipid fractions, coronary artery disease risk, and three blood cell traits is a meaningful finding, but we no longer attempt to quantify this evidence with a p-value. <

Additional comments:

B5R. Figure 2 shows that the confidence interval of the secondary peak does not cover 0. Does that disprove model (6) which says that the invalid instruments are centered around 0?

> The reviewer is correct that the distribution of the estimands from invalid instruments are not centred around zero. However, this distribution is not the focus of estimation. In an earlier version of the method, we did attempt to model the distribution of the estimands from invalid instruments. However, estimating these nuisance parameters did not result in better inferences. Rather it resulted in greater uncertainty as to which variants were invalid instruments.

> We have clarified in the manuscript that we are not concerned about the distribution of estimands from invalid instruments, and we do not attempt to model this distribution: “While it would be possible to estimate the distribution of the invalid estimands, their distribution is not the focus of the investigation. Our attempts to model their distribution resulted in greater uncertainty in which variants were valid and invalid instruments, and less reliable inferences overall.” (page 22). <

B6R. Supplementary Table A1 shows that the coverage of the proposed contamination mixture model is always lower than the nominal 95% level (and may be as low as 30% in unfavorable situations). Why did the authors say "One would expect that if the method were theoretically flawed, that this would manifest in poor performance, which is not the case for the contamination mixture model" in response to my criticism of the theoretical soundness?

> As shown in the additional simulation study (see point B1R), when all genetic variants are valid instrumental variables, coverage under the null is exactly at the nominal 95% level. In the asymptotic sample size limit (that is, an infinite number of individuals in the analysis), invalid instruments should not influence estimates or inferences from the method (under the “plurality valid” assumption). However, with a finite sample size, invalid instrumental variables will still influence the analysis and lead to inflated Type 1 error rates. This is not unique to the contamination mixture method, and occurs for all robust methods for Mendelian randomization (as shown in this manuscript). However, in the simulation study with 20 invalid instruments, Type 1 error rates for the contamination mixture method were still reasonably well controlled (less than 10%). A similar observation was made in a recent comparison of methods (<https://www.biorxiv.org/content/10.1101/577940v1>): the contamination mixture method was the only one of 9 methods to have well-controlled Type 1 error rates across a range of simulation scenarios with up to 50% invalid instruments.

> Referring to reviewer 3’s comment: “I don’t know why in scenario 2, their new method has consistently low coverage, while even IVW method has perfect coverage” – we appreciate the concern over coverage in scenario 2. The explanation is that the IVW method is performed with a random-effects model. It cannot reject outlying genetic variants as invalid, and hence in the presence of balanced zero-mean pleiotropy, the confidence interval for the causal estimate widens, and nominal Type 1 error rates are maintained. However, this is only the case for balanced zero-mean pleiotropy – other types of pleiotropy result in inflated Type 1 error rates. In contrast, the contamination mixture method judges variants with outlying causal estimates to be invalid, and removes them from the analysis – it does the same for all types of pleiotropy. As it is not able to do this perfectly with a finite sample size, it has inflated Type 1 error rates. However, the degree of inflation is not substantial in any scenario.

> In reality, robust methods for Mendelian randomization will only be performed if the standard method for Mendelian randomization (the IVW method) suggests the presence of a causal effect. Hence undercoverage in scenario 2 is less concerning, as in cases where the contamination mixture method incorrectly rejects the null but the IVW method does not reject the null, the contamination mixture method would not have been performed.

> We have clarified in the manuscript the reason why Type 1 error inflation is not observed in the IVW method for Scenario 2: “The IVW method performs well in Scenario 2, as the random-effect model is able to capture balanced pleiotropic effects with mean zero. However, it is unable to model other types of pleiotropy.” (page 6) and “While the

contamination mixture method has slightly inflated Type 1 error rates in Scenario 2, robust methods are typically only used when the standard method (that is, the IVW method) suggests a causal effect. Hence this is unlikely to lead to additional false positive findings in practice.” (page 6). <

B7R. If like the authors suggested in the abstract, that there is a "shared mechanism linking lipids and CHD risk relating to platelet aggregation", why not simply check this by running another MR study using traits indicating platelet aggregation as the exposure?

> While we appreciate the reviewer’s suggestion, the focus of this manuscript is the development of a statistical method – the contamination mixture method. The analysis of HDL-cholesterol and CAD risk is important to demonstrate the utility of the method, but this is not the focus of the manuscript. Performing the reviewer’s proposed analysis would not be a straightforward application of Mendelian randomization – we would have to consider which of the platelet aggregation traits to use as our risk factor, select variants that are associated with this risk factor, and adequately address the question of pleiotropy, as several of the genetic variants in Table 3 are associated with all three blood cell traits.

> In fact, we are currently pursuing this analysis in a separate paper, but using a different methodological strategy (Bayesian model averaging) to perform variable selection on the red blood cell traits. Due to the complexity of this analysis and the lack of direct relevance to the focus of this paper, we believe that the literature would be better served by keeping these pieces of work distinct. <

B8R. If I understand it correctly, the authors' "bootstrap p-value" is just a permutation test. Why is it called bootstrap?

> As per the response to B4R, this approach has been removed from the manuscript. <

Reviewer #3 (Remarks to the Author):

COR. The authors did a good job reassuringly addressing most of my comments. Only three comments require further work in my view.

> We appreciate the reviewer’s perspective and positive comments on the revision. <

C1R. Thanks for these explanations. Could you give clear guidance what to do if different ψ values lead to different causal effect estimates? Supplementary Table A5 goes into this direction, but the results are rather worrying, different ψ lead to vastly different causal effects – it seems we could get any answer we want by setting ψ accordingly. The major claimed insight in the application was the detection of two distinct slopes for the HDL->CHD relationship, but it is supported only for some limited values of ψ . How can we know which one is correct? This makes me think: why not first generate the profile likelihood over a grid of ψ and θ values? This would mean no tuning is needed for ψ .

> As ψ decreases, a variant is only excluded from the analysis if its causal estimate is statistically compatible with a causal effect close to zero. Hence too many variants will be judged as valid, and bimodality may not be detected as the estimates at the two peaks contain too many variants in common.

> As ψ increases, a variant can be excluded from the analysis if its causal estimate is different from the proposed causal effect even if its value is not compatible with zero. However, as ψ increases further, the likelihood function for invalid instruments is spread across a much wider range. It therefore takes lower values across its distribution, making it less likely that variants are judged to be invalid.

> To illustrate, suppose a genetic variant has an estimate of $\hat{\theta}_j = 3$ and a standard error of $se(\hat{\theta}_j) = 0.4$. When comparing to a causal parameter of $\theta = 2$, the value of the likelihood function as a valid instrument is 0.0438. This is unsurprisingly a low value – the variant’s estimate is 2.5 standard errors away from the true causal effect, which would only be expected to occur for 1% of valid instruments. The value of the likelihood function as an invalid instrument depends on the value of ψ :

ψ	Likelihood as an invalid instrument	Judgement
0.5	0.0000	Valid
1	0.0077	Valid
1.5	0.0397	Valid
2	0.0663	Invalid
2.5	0.0781	Invalid
3	0.0807	Invalid
3.5	0.0788	Invalid
4	0.0751	Invalid
4.5	0.0708	Invalid
5	0.0665	Invalid
5.5	0.0624	Invalid
6	0.0586	Invalid
6.5	0.0551	Invalid
7	0.0519	Invalid
7.5	0.049	Invalid
8	0.0464	Invalid
8.5	0.0441	Invalid
9	0.0419	Valid
9.5	0.0399	Valid
10	0.0381	Valid

> We recall that a variant is judged to be valid if the value of the likelihood as a valid instrument (here, 0.0438) is greater than the value as an invalid instrument. The variant is therefore judged to be invalid for values of ψ between 2 and 8.5, and valid for both smaller and larger values.

> The value of the causal estimate depends on which genetic variants are judged to be valid and which to be invalid, and this determination depends on the value of ψ . It is important therefore to take a value of ψ that is not too large or too small, as both these cases make it less likely that a genetic variant having an estimate that is not compatible with the proposed value of the causal effect would be discarded as being invalid.

> A multimodal likelihood is more likely when the set of valid instruments is more sensitive to changes in the value of θ , which again happens when ψ takes a value that is neither too large nor too small.

> We advise that a multimodal distribution should be noted when it occurs for any value of ψ . However, interest in a subgroup of variants will naturally be greater when the subgroup is reasonably large. Subgroups that only contain two or three variants could occur simply due to chance alignment of causal estimates.

> As for estimation, if the causal estimate varies considerably for different values of ψ , this is evidence that no single causal estimate is suggested by the majority of genetic variants. In such a situation, we would advise against giving any single causal estimate based on all the genetic variants, but instead to investigate the source of heterogeneity between the causal estimates (as we have done in this manuscript).

> We have added this advice to the manuscript: “The value of ψ influences which genetic variants are judged to be valid and invalid. Variants are less likely to be judged to be invalid if ψ is too large or too small. If multimodality in the likelihood is detected for any value of ψ , then this can be interpreted as evidence for the presence of multiple causal mechanisms. If the causal estimate varies considerably for different values of ψ , this suggests that not all genetic variants are valid instruments, and researchers are discouraged from presenting any of these estimates as a single definitive causal estimate. Instead, we encourage researchers to consider whether some variants should be removed from the analysis for being pleiotropic, or to find clusters of variants with similar causal estimates that may represent a coherent causal mechanism.” (page 23).

C6R. Thanks for the clarification around the applied example. This allowed me to digest these results. I am still puzzled for several reasons:

C6Ra. The authors talk about two local maxima in the profile likelihood 0.67 and 0.93, but the slopes in the Figure are negative and of course the causal effect should be negative. Is there a typo?

> 0.67 and 0.93 are causal estimates on the odds ratio scale. An odds ratio less than 1 suggests a protective effect of a risk factor on an outcome. The corresponding log odds ratios are negative. The y-axis in the Figure is the log-odds ratio. Hence the slopes are negative. This has now been clarified in the manuscript: “Estimates are less than one, suggesting a protective effect of HDL-cholesterol on CHD risk.” (page 6). <

C6Rb. The definition of valid instrument is quite arbitrary. If for example the true causal effect is zero and there are a set of SNPs showing correlated pleiotropy [<https://www.biorxiv.org/content/10.1101/682237v1>] for another trait, wouldn't the method pick up those pleiotropic SNP as the most likely valid instruments? In such scenario does it make any sense to look for enrichment for trait-associated SNPs among the more valid instruments?

> “Correlated pleiotropy” is a term recently coined for the scenario in which the primary risk factor influenced by the genetic variant has been incorrectly identified, and the genetic variant in fact directly influences a precursor of the risk factor. The genetic association with the risk factor is mediated entirely via its precursor. In the scenario that the reviewer describes, the contamination mixture method would likely identify two subsets of variants: those that influence the risk factor directly, and those that primarily influence the precursor of the risk factor. The causal estimate from the contamination mixture method would correspond to which of these subsets corresponded to a more precise causal effect estimate. If the two subsets had similar strength of evidence, then the confidence interval would comprise two intervals corresponding to the distinct subsets.

> In this case, an enrichment analysis would reveal that one subset of variants is associated with the precursor of the risk factor and any other variables downstream of this, and the other subset is not. This would help the investigator understand why two subsets are identified, and would help identify the causal mechanism (whether it operated via the risk factor, or via its precursor). If the subset of variants associated with the precursor suggested

a causal effect, but the subset not associated with the precursor did not suggest a causal effect, this would suggest that it is the precursor that is the causal risk factor, and that the risk factor itself is not a causal risk factor. This is a clear example where the contamination mixture method is worthwhile and can provide useful insights into causal mechanisms.

> We have added to the manuscript: “Identifying variables that associate with a cluster of variants may help us identify mediators on a particular causal mechanism, or it may help us identify a precursor of the nominal risk factor that is the true causal factor.” (page 11) <

C6Rc. I still struggle to understand what the enrichment analysis shows. Despite my concern (see point b) about the meaning of “valid instrument” indicated by your method, I wonder what it means if many of those SNPs that tend to be more valid are enriched to be associated with another trait? Is this other trait (e.g. PDW) a mediator or rather a trait for which these “valid instruments” may exhibit pleiotropy? The latter one would exactly mean that they are invalid, so I assume you rather mean that these traits are mediators or what else could they be? Why invalid instruments should be less enriched for a potential mediator? Why is the path through “valid instruments” more meaningful than the path through “invalid instruments” (which are mostly invalid because of pleiotropy, which is biologically just as meaningful)?

> There are a number of ways that one could interpret the presence of multiple mechanisms linking a risk factor to an outcome. It could be that the risk factor is not truly a single entity – for example, serum cholesterol is not a single entity, but a combination of HDL-cholesterol, LDL-cholesterol, IDL-cholesterol, and so on – and different aspects of the risk factor influence the outcome in different ways. Or it could be that the risk factor is a single entity, but there are different ways of intervening on the risk factor – for example, one could intervene on BMI by increasing physical activity, or by reducing caloric intake. Or it could be that the explanation is mediation, as suggested by the reviewer, and different variants influence the outcome via different mediated pathways. Or it could be that the “correlated pleiotropy” explanation is true, and some of the genetic variants primarily influence a precursor of the risk factor (which could be the variable identified by the enrichment analysis). The contamination mixture method is agnostic to how different causal effects occur. The method simply provides evidence for the presence of multiple distinct causal effects. It is not able to distinguish between these scenarios – that requires biological knowledge, and cannot be done using statistical methods alone.

> In our example, we focused on the more negative causal estimate as this was the estimate which the greatest evidence as judged by the contamination mixture method, and because the variants that evidenced this causal effect were more distinct than those evidencing the less negative estimate.

> One potential explanation of the findings is that there is some latent precursor of HDL-cholesterol and these blood cell traits, and the variants strongly associated with CHD risk are the variants that influence this latent precursor. In contrast, variants that influence HDL-cholesterol directly are less strongly associated with CHD risk. However, we want to be cautious not to be overly speculative – it may be that one of the other explanations listed above may be true.

> We have added to the manuscript: “There are many reasons why multiple distinct causal effect estimates may be evidenced. It may be that the risk factor is not a single entity, but a compound measurement incorporating multiple risk factors with different causal effects. It may be that the risk factor is a single entity, but there are different ways to intervene on it,

leading to different magnitudes of causal effects. Alternatively, it may be that some variants do not affect the risk factor directly, but rather a precursor of the risk factor.” (page 10-11).

<

> When investigating the presence of multiple causal mechanisms, we would not want to call one subset of variants “valid” and others “invalid”. This is now clarified in the manuscript: “When multiple causal effects are evidenced in the data, we would not want to label one subset of genetic variants as ‘valid’ and others as ‘invalid’ in an absolute sense. Validity of variants is relative to the proposed value of the causal effect – different subsets of variants will be valid for different values of the causal effect.” (page 11). <

C6Rd. What is described here “We investigated variants that were associated with increased HDL-cholesterol, decreased CHD risk, and at least one of the above blood cell traits” and below has nothing to do with the findings of the proposed method and comes a bit out of the blue to me.

> The contamination mixture method identifies clusters of variants with similar causal estimates. The motivation for the enrichment analysis was to assess whether variants in this cluster had any commonality, which may give clues to identify the causal mechanism by which these variants (but not others) were associated with the outcome. This is now clearly stated in the manuscript: “We proceed to investigate whether there are any traits that preferentially show associations with genetic variants in this cluster as opposed to with variants not in this cluster, as this may help us identify the mechanism driving the genetic associations with lower CHD risk.” (page 7). <

C6Re. It should be noted that only PDW survives Bonferroni correction, the other traits could occur by chance (or by being correlated to PDW).

> As stated in the response to point B4R, this analysis has been removed from the paper. <

C6Rf. In general, I believe that for the readers it would be extremely useful to show a DAG (including HDL, PDW, CHD and potentially other factors) describing how the presumably valid instruments (higher posterior probability of being valid) are enriched for being associated with platelet distribution width. Testing enrichment without any particular underlying model in mind is less meaningful.

> While we agree that this would be useful, it would also be entirely speculative. As per the response to C6Rc, our method is agnostic to the mechanism by which the different causal effects occur, and does not provide any clue as to how the variables are related. We agree that hypothesis-free searches for mechanisms are less reliable than hypothesis-driven investigations, and would regard analyses like the one in the paper are hypothesis-generating. However, they are useful either to suggest novel biological pathways, or when there is prior epidemiological knowledge, as in the example of BMI and Type 2 diabetes.

> We have stated clearly in the paper: “However, our investigation only highlights these traits as associated with variants in the cluster of variants having negative associations with CHD risk. It does not give any further indication of how the mechanism operates.” (page 9). <

C8R. The fact that (under some ψ settings) the data yields evidence for two distinct slopes does not mean that there is no single causal effect, it simply means that there are SNPs that violate an MR assumption (most probably the exchangeability, see “correlated pleiotropy” above). Of course, it could also mean context-dependent causal effects, but the marginal causal effect (integrating out all

other factors) is unique. Moreover, stratified mediation analysis could be done by using two groups of instruments, once those supporting the 0.67 causal effect, once the other SNPs leading to 0.93 causal effect.

> We do not see any strong justification for interest in the marginal causal effect when there is evidence in the data for multiple causal effects. The marginal causal effect does not represent a causal effect relating to any particular mechanism, but rather an average across mechanisms.

> It is possible that the variants in the cluster are all pleiotropic. However, if they are all pleiotropic in the same way, then this represents a causal mechanism influencing the outcome. It is also possible that they are all pleiotropic in different ways and it is purely coincidence that the causal effect estimates are all similar. However, when there is a large cluster of variants all evidencing the same causal effect on the outcome, it is increasingly unlikely that this is purely a coincidental finding. We now discuss this in the manuscript: "A further possibility is that the link with the variable is coincidental. However, if multiple variants in the cluster are all associated with the same variable with the same direction of association, then a common mechanism is likely." (page 11).

> As per the response to C6Rf, a mediation analysis would only be valid under the assumption that the effect of HDL-cholesterol is mediated via one or other of the blood cell traits. Performing such an analysis would be overly speculative, as there are many other reasons why multiple mechanisms may be observed. In particular, the "correlated pleiotropy" scenario may be the truth, in which case the blood cell traits are a precursor of HDL-cholesterol, and not a mediator. <

Comments from reviewer 3 on reviewer 2's comments:

C9R. In summary, while reviewer #2 claims that the method is flawed, in my view it is not fundamentally flawed (apart from the estimation of the confidence interval, which may be incorrect), but hinges on classical MR assumptions (as [almost] all MR methods). Still it needs some clarifications and importantly a convincing argument (implying new analysis) why the Wilks theorem guarantee normality and why it provides incorrect coverage for easy scenarios (#2: 83.6-93.1 instead of 95). I do not see this a major hurdle, but more work is needed.

> We have now demonstrated that the Wilks' theorem provides correct coverage properties in a range of scenarios when the instrumental variable assumptions are satisfied, even when the number of genetic variants is low and the number of participants in the analysis is moderate (see point B1R). <

C10R. I maintain my comment that the major weakness of the method is that by tuning parameter ψ , one can obtain almost arbitrary causal effect estimates - which is a serious problem, but (as far as I can see) could be solved by using a profile likelihood on a grid both on ψ and θ .

> As per the response to C1R, if different values of ψ lead to substantially different causal estimates, then the genetic variants are not providing strong evidence of a single causal estimate. Rather than a weakness, we would regard being able to detect such cases as a strength of the method. In contrast, all other robust methods would not detect such inconsistencies in the analysis.

> It is not clear that joint maximization of ψ and θ would be worthwhile, as ψ is a nuisance parameter. Estimating the value of ψ could lead to increased sensitivity to variants to outlying causal estimates, as such variants would influence the value of ψ . <

C11R. My comments regarding the application reflect the fact that I am not sure of the meaning of their finding, no interpretation is offered or tested (while could have been done). Furthermore, I do not see much point in the subgroup membership prediction exercise (no clear hypothesis is formulated, no DAG is drawn to present it, etc.).

> As we have stated in the abstract, the finding is that there is evidence for multiple clusters of HDL-cholesterol associated variants having distinct associations with coronary heart disease risk, and for one of these clusters, we demonstrated “11 variants associated with increased HDL-cholesterol, decreased triglyceride levels, and decreased CHD risk that had the same directions of associations with platelet distribution width and other blood cell traits, suggesting a shared mechanism linking lipids and CHD risk relating to platelet aggregation” (page 2).

> As we have stated in the response (points C6Rf and C8R), this a hypothesis-generating approach that highlights potential biological mechanisms. It does not provide definitive biological knowledge as to how variables are related. This is now clarified in the manuscript: “We note that this investigation is performed without a prior hypothesis, and so should be regarded as an exploratory ‘hypothesis-generating’ investigation.” (page 7-8). <

Reviewers' Comments:

Reviewer #3:

Remarks to the Author:

I thank the authors for their considerable effort to improve the m/s based on the formulated criticism. I am convinced now of almost all of the remaining aspects I flagged up, but find it very surprising how hesitant they are to perform some minor follow-up analyses in order to generate (or at least list) some testable concrete hypotheses instead of an extremely vague biological conclusion. Proposing some explanations and perform some preliminary testing with very cautious interpretation would make the paper much more interesting to the readership of Nat comms.

C1R "Variants are less likely to be judged to be invalid if ψ is too large or too small. If multimodality in the likelihood is detected for any value of ψ , then this can be interpreted as evidence for the presence of multiple causal mechanisms. If the causal estimate varies considerably for different values of ψ "

R-C1R: I was expecting to have more quantitative recommendations than "too large or too small" and "varies considerably", but I appreciate that the authors admit that this is as good as it can get.

C6Rb. "In the scenario that the reviewer describes, the contamination mixture method would likely identify two subsets of variants: those that influence the risk factor directly, and those that primarily influence the precursor of the risk factor."

R - C6Rb. I agree that two sets would be defined, but the ones termed as "valid instruments" would actually be the invalid ones (the indirect effects are compatible with a non-zero slope (LVj), which the valid ones would be concordant with the mixture component with zero mean (LFj). Later the authors note that "we would not want to label one subset of genetic variants as 'valid' and others as 'invalid' in an absolute sense."

C6Rc. "It could be that the risk factor is not truly a single entity – for example, serum cholesterol is not a single entity, but a combination of HDL-cholesterol, LDL-cholesterol, IDL-cholesterol, and so on – and different aspects of the risk factor influence the outcome in different ways. Or it could be that the risk factor is a single entity, but there are different ways of intervening on the risk factor – for example, one could intervene on BMI by increasing physical activity, or by reducing caloric intake. Or it could be that the explanation is mediation, as suggested by the reviewer, and different variants influence the outcome via different mediated pathways. Or it could be that the "correlated pleiotropy" explanation is true, and some of the genetic variants primarily influence a precursor of the risk factor (which could be the variable identified by the enrichment analysis)."

"One potential explanation of the findings is that there is some latent precursor of HDL cholesterol and these blood cell traits, and the variants strongly associated with CHD risk are the variants that influence this latent precursor. In contrast, variants that influence HDL cholesterol directly are less strongly associated with CHD risk"

R-C6Rc. Listing such explanations in the manuscript would be extremely useful, instead of being overly cautious and not to formulate any potential explanation.

C6Rf. "our method is agnostic to the mechanism by which the different causal effects occur" and "would regard analyses like the one in the paper as hypothesis generating", "only highlights these traits as associated with variants"

R-C6Rf. My problem is that there is no testable hypothesis generated here. Some causal effect heterogeneity is detected, so there may be multiple causal mechanisms, which are somehow

linked with platelet aggregation. Such a statement cannot be tested – it leads to no experimental design that could answer this. What would be meaningful is at least propose a few testable hypotheses how platelet aggregation may be implicated (even if those are speculative, but at least concrete testable hypotheses (often described by DAGs). Otherwise what is the point of doing all the follow-up analysis to point to platelet aggregation (which is the main biological insight of the paper) if it only leads to such an extremely vague statement?

C8R. "...a mediation analysis would only be valid under the assumption that the effect of HDL-cholesterol is mediated via one or other of the blood cell traits. Performing such an analysis would be overly speculative, as there are many other reasons why multiple mechanisms may be observed. "

R-C8R. In my view, the sentence in the abstract

"In a MR analysis for high-density lipoprotein (HDL) cholesterol and coronary heart disease (CHD) risk, the method identified 11 variants associated with increased HDL-cholesterol, decreased triglyceride levels, and decreased CHD risk that had the same directions of associations with platelet distribution width and other blood cell traits, suggesting a shared mechanism linking lipids and CHD risk relating to platelet aggregation."

remains entirely speculative if the authors do not make the slightest effort to at least try any of the proposed analysis. I cannot reasonably understand the authors' reluctance to at least attempt a stratified mediation analysis. If the answer is non-conclusive, it can be admitted. If it shows some promising trend that some instruments favour platelet aggregation as potential mediator/precursor (for example) it is a plus to mention, even if it still remains speculative, but less speculative than a sentence (cited above) that is based on little evidence without any attempt of any consolidation whatsoever. Also, "Performing such an analysis would be overly speculative", performing such an analysis is NOT speculative, but to overinterpreting the results of such analysis could be speculative.

C10R "It is not clear that joint maximization of ψ and θ would be worthwhile, as ψ is a nuisance parameter. Estimating the value of ψ could lead to increased sensitivity to variants to outlying causal estimates, as such variants would influence the value of ψ ."

R -C10R. I do not feel this argument compelling. The authors (again) could have easily tested this at least for one example/simulation setting (with outlier pleiotropic effect) to see if it confirms their hunch feeling.

Reviewer #3 (Remarks to the Author):

I thank the authors for their considerable effort to improve the m/s based on the formulated criticism. I am convinced now of almost all of the remaining aspects I flagged up, but find it very surprising how hesitant they are to perform some minor follow-up analyses in order to generate (or at least list) some testable concrete hypotheses instead of an extremely vague biological conclusion. Proposing some explanations and perform some preliminary testing with very cautious interpretation would make the paper much more interesting to the readership of Nat comms.

> We thank the reviewer for their positive comments. Our reasoning for not performing the requested analyses in the previous draft was that we were being cautious about analyses performed after a hypothesis-generating investigation. We do appreciate the reviewer's assertion that this would provide a less vague overall conclusion and so have now performed the suggested analyses which provide a more concrete interpretation of the results.

> We would contest the reviewer's assertion that no hypotheses are tested in the manuscript (see point C6Rf below). But the reviewer is correct that we previously left the conclusion ambiguous rather performing additional analyses. This has now been put right. <

C6Rb. "In the scenario that the reviewer describes, the contamination mixture method would likely identify two subsets of variants: those that influence the risk factor directly, and those that primarily influence the precursor of the risk factor."

R - C6Rb. I agree that two sets would be defined, but the ones termed as "valid instruments" would actually be the invalid ones (the indirect effects are compatible with a non-zero slope (LVj)), which the valid ones would be concordant with the mixture component with zero mean (LFj). Later the authors note that "we would not want to label one subset of genetic variants as 'valid' and others as 'invalid' in an absolute sense."

> When the aim of the investigation is to identify distinct subgroups of genetic variants having mutually similar causal estimates, if multiple subsets of variants are identified by the method, we would not want to label one particular cluster as valid and the other as invalid. Different clusters represent different causal mechanisms linking the risk factor to the outcome. Analyses such as the one we performed in this paper could identify traits associated with variants in each cluster, providing evidence on the causal mechanism for each cluster.

> We have added a sentence stating: "Any cluster of variants may indicate a causal mechanism linking the risk factor to the outcome, even if it is not the largest cluster." (page 11). This clarifies the previous sentence that "we would not want to label one subset of genetic variants as 'valid' and others as 'invalid' in an absolute sense" (page 11). <

C6Rc. "It could be that the risk factor is not truly a single entity – for example, serum cholesterol is not a single entity, but a combination of HDL-cholesterol, LDL-cholesterol, IDL-cholesterol, and so on – and different aspects of the risk factor influence the outcome in different ways. Or it could be that the risk factor is a single entity, but there are different ways of intervening on the risk factor – for example, one could intervene on BMI by increasing physical activity, or by reducing caloric intake. Or it could be that the explanation is mediation, as suggested by the reviewer, and different variants influence the outcome via different mediated pathways. Or it could be that the "correlated pleiotropy" explanation is true, and some of the genetic variants primarily influence a precursor of the risk factor (which could be the variable identified by the enrichment analysis)."

“One potential explanation of the findings is that there is some latent precursor of HDL cholesterol and these blood cell traits, and the variants strongly associated with CHD risk are the variants that influence this latent precursor. In contrast, variants that influence HDL cholesterol directly are less strongly associated with CHD risk”

R-C6Rc. Listing such explanations in the manuscript would be extremely useful, instead of being overly cautious and not to formulate any potential explanation.

> We had previously included these explanations in a general context in the Discussion section of the manuscript (page 11). We have now added them to the Results section in the specific context of the HDL-cholesterol analysis:

“There are several potential reasons why multiple magnitudes of causal effect may be evidenced in the data (Supplementary Figure A5). HDL-cholesterol is not a single entity. It could be that different size categories of HDL particles influence CHD risk to varying extents. Alternatively, it may be that there are multiple mechanisms by which HDL-cholesterol influences CHD risk, and different genetic variants act as proxies for distinct mechanisms. The identified blood cell traits may act as mediators on one or other of these pathways. Or it could be that the traits are precursors of the risk factor, and some variants influence the traits rather than HDL-cholesterol directly.” (page 9).

> We have also added Supplementary Figure A5, which illustrates these possibilities graphically:

C6Rf. “our method is agnostic to the mechanism by which the different causal effects occur” and “would regard analyses like the one in the paper are hypothesis generating”, “only highlights these traits as associated with variants”

R-C6Rf. My problem is that there is no testable hypothesis generated here. Some causal effect heterogeneity is detected, so there may be multiple causal mechanisms, which are somehow linked with platelet aggregation. Such a statement cannot be tested – it leads to no experimental design that could answer this. What would be meaningful is at least propose a few testable hypotheses how

platelet aggregation may be implicated (even is those are speculative, but at least concrete testable hypotheses (often described by DAGs). Otherwise what is the point of doing all the follow-up analysis to point to platelet aggregation (which is the main biological insight of the paper) if it only leads to such an extremely vague statement?

> Our testable hypothesis is that the genetic variants in the cluster are more associated with one or more traits in PhenoScanner than the variants that are not in the cluster. The null hypothesis is that all traits are equally associated with variants in the cluster as with variants not in the cluster. We found 3 traits that were preferentially associated with variants in the cluster. In a previous version of the manuscript, we provided a bootstrapped p-value as an indication of the strength of evidence for this hypothesis. However, assessing the degree of multiple testing for this hypothesis was subjective, and the p-value did not take into account that all 11 variants had the same direction of association with the top 3 traits. Hence we omitted these p-values from the updated version of the manuscript.

> The colocalization analyses provide another testable hypothesis. In this case, the null hypothesis is that the blood cell traits do not colocalize with HDL-cholesterol and/or CHD risk. However, in 7 of the 9 gene regions, evidence for colocalization was observed.

> However, we take the reviewer’s point that we could explore the relationships between the variables in a more quantitative way. We therefore performed a mediation analysis using the 11 variants in this cluster. We adjusted for genetic associations with each of the blood cell traits in turn using multivariable Mendelian randomization. Results are shown below and in Supplementary Table A6:

	Estimate for HDL-cholesterol	95% CI
No adjustment	-0.584	-0.828, -0.340
Adjustment for MCHC	-0.238	-0.651, 0.175
Adjustment for PDW	-0.381	-0.931, 0.169
Adjustment for RCDW	-0.275	-0.759, 0.210

> We see that the coefficient for HDL-cholesterol attenuates substantially on adjustment for each of the traits, suggesting that the causal effect may be mediated in part via these blood cell traits, and by mean corpuscular haemoglobin concentration in particular.

> In contrast, if we adjust for alternative cardiovascular risk factors (as negative controls): BMI (genetic associations estimated in the GIANT consortium 2015 data release), systolic blood pressure (estimated in the UK Biobank, 2019 Ben Neale data release), or Type 2 diabetes (estimated in the DIAGRAM consortium, 2017 data release), associations did not attenuate substantially:

	Estimate for HDL-cholesterol	95% CI
No adjustment	-0.584	-0.828, -0.340
Adjustment for SBP	-0.477	-0.646, -0.307
Adjustment for BMI	-0.487	-0.695, -0.279
Adjustment for T2D	-0.545	-0.888, -0.201

> We have added the mediation analysis to the manuscript: “To assess this, we performed a mediation analysis in which we adjusted for genetic associations with each of the blood cell traits in turn using multivariable Mendelian randomization. The coefficient for the causal

effect of HDL-cholesterol attenuated substantially on adjustment for each of the traits (Supplementary Table A6), and particularly on adjustment for mean corpuscular haemoglobin concentration, suggesting that at least part of the causal effect may be mediated via these blood cell traits. In contrast, associations did not attenuate substantially on adjustment for alternative cardiovascular risk factors (Supplementary Table A6).” (page 9). <

C8R. “...a mediation analysis would only be valid under the assumption that the effect of HDL-cholesterol is mediated via one or other of the blood cell traits. Performing such an analysis would be overly speculative, as there are many other reasons why multiple mechanisms may be observed.”

R-C8R. In my view, the sentence in the abstract

“In a MR analysis for high-density lipoprotein (HDL) cholesterol and coronary heart disease (CHD) risk, the method identified 11 variants associated with increased HDL-cholesterol, decreased triglyceride levels, and decreased CHD risk that had the same directions of associations with platelet distribution width and other blood cell traits, suggesting a shared mechanism linking lipids and CHD risk relating to platelet aggregation.”

remains entirely speculative if the authors do not make the slightest effort to at least try any of the proposed analysis. I cannot reasonably understand the authors’ reluctance to at least attempt a stratified mediation analysis. If the answer is non-conclusive, it can be admitted. If it shows some promising trend that some instruments favour platelet aggregation as potential mediator/precursor (for example) it is a plus to mention, even if it still remains speculative, but less speculative than a sentence (cited above) that is based on little evidence without any attempt of any consolidation whatsoever. Also, “Performing such an analysis would be overly speculative”, performing such an analysis is NOT speculative, but to overinterpreting the results of such analysis could be speculative.

> We have added the mediation analysis to the manuscript, which does indeed point to the blood cell traits as potential mediators of the effect of HDL on CHD. <

C10R “It is not clear that joint maximization of ψ and θ would be worthwhile, as ψ is a nuisance parameter. Estimating the value of ψ could lead to increased sensitivity to variants to outlying causal estimates, as such variants would influence the value of ψ .”

R -C10R. I do not feel this argument compelling. The authors (again) could have easily tested this at least for one example/simulation setting (with outlier pleiotropic effect) to see if it confirms their hunch feeling.

> In the applied example of HDL-cholesterol on CHD risk, the value of ψ that maximizes the log-likelihood is 0.29. In contrast, our recommended initial value of ψ is 1.11. These different values lead to substantially different inferences. A bimodal distribution is still observed with $\psi = 0.29$, although the lower peak no longer contributes to the 95% confidence interval. However, in a single example, it is not clear whether this result is more or less reliable than the result from the original version of the method.

> We re-ran the simulation study for 1000 iterations per invalid instrument scenario with a null causal effect, implementing the original version of the method, and the proposed version maximizing the log-likelihood with respect to ψ and θ . Results were as follows:

	20 invalid			40 invalid			60 invalid		
	Mean	SD	Type 1 error	Mean	SD	Type 1 error	Mean	SD	Type I error
Scenario 2: Balanced pleiotropy, InSIDE satisfied									
Original	0.001	0.033	6.5	0.001	0.043	9.4	0.001	0.065	15.6
Proposed	0.005	0.080	7.9	0.002	0.060	9.9	0.002	0.070	16.2
Scenario 3: Directional pleiotropy, InSIDE satisfied									
Original	0.015	0.033	8.8	0.041	0.045	26.4	0.139	0.164	61.7
Proposed	0.606	0.418	72.5	0.740	0.377	87.4	0.770	0.362	95.6
Scenario 4: Directional pleiotropy, InSIDE violated									
Original	0.011	0.035	9.2	0.034	0.055	18.8	0.161	0.159	52.4
Proposed	0.113	0.204	29.3	0.079	0.144	33.0	0.155	0.138	61.9

> The slight differences from results in the paper for the original version of the method arise because only 1000 iterations were performed. This table is included in the manuscript as Supplementary Table A8.

> We see that the proposed approach performs less well than the original approach in terms of bias, efficiency, and coverage. Differences in Scenario 3 are particularly striking. By checking some specific example datasets, the proposed method often selects a value of ψ that leads to a large group of variants with less similar causal estimates being included in the analysis as valid instruments. This compares with the original version, which excludes more variants from the analysis, resulting in a lower likelihood, but more robust inferences.

> We have added: “As an alternative approach, we considered joint maximization of the likelihood across both ψ and θ . We re-ran the simulation study for 1000 iterations per invalid instrument scenario with a null causal effect, implementing the original version of the method, and the proposed version jointly maximizing the log-likelihood with respect to ψ and θ . Results are shown in Supplementary Table A8 We see that the joint maximization approach performs less well than the original approach in terms of bias, efficiency, and Type 1 error rate (particularly in Scenario 3). By checking some specific example datasets, we noted that the joint maximization version often selects a value of ψ that leads to a large group of variants with less similar causal estimates being included in the analysis as valid instruments. This compares with the original version, which excludes more variants from the analysis in these cases, resulting in a lower likelihood, but more robust inferences.” (page 23). <

Reviewers' Comments:

Reviewer #3:

Remarks to the Author:

I sincerely commend the authors for their effort to address my points, they did a great job! I am happy with the additional results (which I believe make the paper stronger) and convinced of the usefulness of their method.

I only have one small comment: Given the additional mediation analysis results I'd update the abstract replacing the statement

"suggesting a shared mechanism linking lipids and CHD risk relating to platelet aggregation"

with a sentence referring to the mediation analysis and its suggestive conclusion that blood cell traits may be potential mediators of the effect of HDL on CHD.

REVIEWERS' COMMENTS:

Reviewer #3 (Remarks to the Author):

I sincerely commend the authors for their effort to address my points, they did a great job! I am happy with the additional results (which I believe make the paper stronger) and convinced of the usefulness of their method.

I only have one small comment: Given the additional mediation analysis results I'd update the abstract replacing the statement

"suggesting a shared mechanism linking lipids and CHD risk relating to platelet aggregation"

with a sentence referring to the mediation analysis and its suggestive conclusion that blood cell traits may be potential mediators of the effect of HDL on CHD.

> We have changed this phrase to: "suggesting a shared mechanism linking lipids and coronary heart disease risk mediated via platelet aggregation." <